# Modeling and mechanical perturbations reveal how spatially regulated anchorage gives rise to spatially distinct mechanics across the mammalian spindle

**Pooja Suresh[1,2†], Vahe Galstyan[3,4†], Rob Phillips[5,6,7]\*, Sophie Dumont[1,2,7,8]\***

[1]Biophysics Graduate Program, University of California, San Francisco, San Francisco, United States; [2]Department of Bioengineering and Therapeutic Sciences, University of California, San Francisco, San Francisco, United States; [3]Biochemistry and Molecular Biophysics Option, California Institute of Technology, Pasadena, United States; [4]A. Alikhanyan National Laboratory (Yerevan Physics Institute), Yerevan, Armenia; [5]Division of Biology and Biological Engineering, California Institute of Technology, Pasadena, United States; [6]Department of Physics, California Institute of Technology, Pasadena, United States; [7]Chan Zuckerberg Biohub, San Francisco, San Francisco, United States; [8]Department of Biochemistry and Biophysics, University of California, San Francisco, San Francisco, United States

**\*For correspondence:**
phillips@pboc.caltech.edu (RP);
sophie.dumont@ucsf.edu (SD)

[†]These authors contributed equally to this work

**Competing interest:** The authors declare that no competing interests exist.

**Abstract** During cell division, the spindle generates force to move chromosomes. In mammals, microtubule bundles called kinetochore-fibers (k-fibers) attach to and segregate chromosomes. To do so, k-fibers must be robustly anchored to the dynamic spindle. We previously developed microneedle manipulation to mechanically challenge k-fiber anchorage, and observed spatially distinct response features revealing the presence of heterogeneous anchorage (Suresh et al., 2020). How anchorage is precisely spatially regulated, and what forces are necessary and sufficient to recapitulate the k-fiber's response to force remain unclear. Here, we develop a coarse-grained k-fiber model and combine with manipulation experiments to infer underlying anchorage using shape analysis. By systematically testing different anchorage schemes, we find that forces solely at k-fiber ends are sufficient to recapitulate unmanipulated k-fiber shapes, but not manipulated ones for which lateral anchorage over a 3 μm length scale near chromosomes is also essential. Such anchorage robustly preserves k-fiber orientation near chromosomes while allowing pivoting around poles. Anchorage over a shorter length scale cannot robustly restrict pivoting near chromosomes, while anchorage throughout the spindle obstructs pivoting at poles. Together, this work reveals how spatially regulated anchorage gives rise to spatially distinct mechanics in the mammalian spindle, which we propose are key for function.

## Editor's evaluation

In this elegant and technically sophisticated study, the authors study the mechanical properties of the mitotic spindle by combining various experimental biophysical approaches, including microneedle manipulation and quantitative imaging, with theoretical modeling. By systematically exploring shapes of unmanipulated and manipulated kinetochore fibers, they provide compelling evidence for a lateral anchor near the chromosomes. These important findings further our understanding of the balance of forces in the entire mitotic spindle. The work should appeal broadly to cell biologists and biophysicists who are interested in the cytoskeleton and cell division.

## Introduction

Cell division is essential to all life. The accurate segregation of chromosomes during cell division is achieved by the spindle, a macromolecular machine that distributes chromosomes equally to each new daughter cell. To perform this mechanical task, the spindle must be dynamic yet structurally robust: it must remodel itself and be flexible, yet robustly generate and respond to force to move chromosomes and maintain its mechanical integrity. How this is achieved remains an open question. Indeed, while much is known about the architecture (*McDonald et al., 1992*; *Mastronarde et al., 1993*) and dynamics (*Mitchison, 1989*) of the mammalian spindle, and the molecules essential to its function (*Hutchins et al., 2010*; *Neumann et al., 2010*), our understanding of how they collectively give rise to robust mechanics and function lags behind.

In the mammalian spindle, kinetochore-fibers (k-fibers) are bundles of microtubules (*McDonald et al., 1992*; *O'Toole et al., 2020*, *Kiewisz et al., 2021*) that connect chromosomes to spindles poles, ultimately moving chromosomes to poles and future daughter cells. To do so, k-fibers must maintain their connection to the dynamic spindle. The k-fiber's connection (anchorage) to the spindle is mediated by a dense mesh-like network of non-kinetochore microtubules (non-kMTs) which connect to k-fibers along their length (*Mastronarde et al., 1993*; *O'Toole et al., 2020*) via both motor and non-motor proteins. Although this network cannot be easily visualized with light microscopy, physical perturbations such as laser ablation (*Kajtez et al., 2016*; *Milas and Tolić, 2016*; *Elting et al., 2017*) and cell compression (*Trupinić et al., 2022*; *Neahring et al., 2021*) have been instrumental in uncovering how this network anchors k-fibers. The non-kMT network bears mechanical load locally (*Milas and Tolić, 2016*; *Elting et al., 2017*), links sister k-fibers together (*Kajtez et al., 2016*), and contributes to k-fiber and spindle chirality (*Trupinić et al., 2022*; *Neahring et al., 2021*). Recent advances in microneedle manipulation enabled us to mechanically challenge k-fiber anchorage with unprecedented spatiotemporal control (*Long et al., 2020*; *Suresh et al., 2020*). Exerting forces at different locations along the k-fiber's length revealed that anchorage is heterogeneous along the k-fiber: k-fibers were restricted from pivoting near kinetochores, mediated by the microtubule cross-linker PRC1, but not near poles (*Suresh et al., 2020*). Such reinforcement helps robustly preserve the k-fiber's orientation in the spindle center, which we speculate forces sister k-fibers to be parallel and promotes correct attachment. However, how this reinforcement is enacted over space, namely how local or global it is, remains unclear. Furthermore, we do not yet understand which connections along the k-fiber's length are necessary and sufficient to give rise to such spatially distinct mechanics.

The precise spatiotemporal control achieved by microneedle manipulation offers rich quantitative information on the k-fiber's anchorage in the spindle (*Suresh et al., 2020*) and demands a quantitative model-building approach for its full interpretation. Knowledge of the spindle connections from electron microscopy (*McDonald et al., 1992*; *Mastronarde et al., 1993*; *O'Toole et al., 2020*) is not sufficient to understand how they collectively reinforce the k-fiber, and perturbing different regions of the network to experimentally test their contribution is challenging. Furthermore, while we can deplete spindle crosslinkers, quantitatively controlling their combined mechanical function over space is not currently within reach. In turn, a coarse-grained modeling approach (accounting for the effective influence of collective molecular actions) can allow us to systematically dissect the spatial regulation of k-fiber anchorage in the spindle. Since the bending mechanics of microtubules is well characterized (*Gittes et al., 1993*), many modeling studies have used shape to infer forces exerted on microtubules. This approach has been applied to single microtubules (*Gittes et al., 1996*; *Brangwynne et al., 2006*), microtubule bundles (*Gadêlha et al., 2013*; *Portran et al., 2013*), as well as k-fibers in the spindle (*Rubinstein et al., 2009*; *Kajtez et al., 2016*). To date, k-fiber models used native shapes (in unperturbed spindles) to infer underlying spindle forces, without focusing on k-fiber anchorage. This is mainly because the presence of anchorage is not easily revealed in unperturbed spindles. Using k-fiber manipulation in mammalian spindles, we are uniquely positioned to probe k-fiber anchorage forces previously hard to detect, and to test models for their underlying basis.

Here, we use coarse-grained modeling and microneedle manipulation experiments to define the spindle anchorage forces necessary and sufficient for the k-fiber to robustly restrict pivoting near kinetochores while allowing pivoting at poles (*Figure 1*, top). We model the k-fiber using Euler-Bernoulli beam theory. We systematically increase model complexity and use shape analysis to infer the minimal set of forces needed to recapitulate experimental k-fiber shapes. We find that while forces and moments at k-fiber ends (end-point anchorage) alone are sufficient to recapitulate unmanipulated

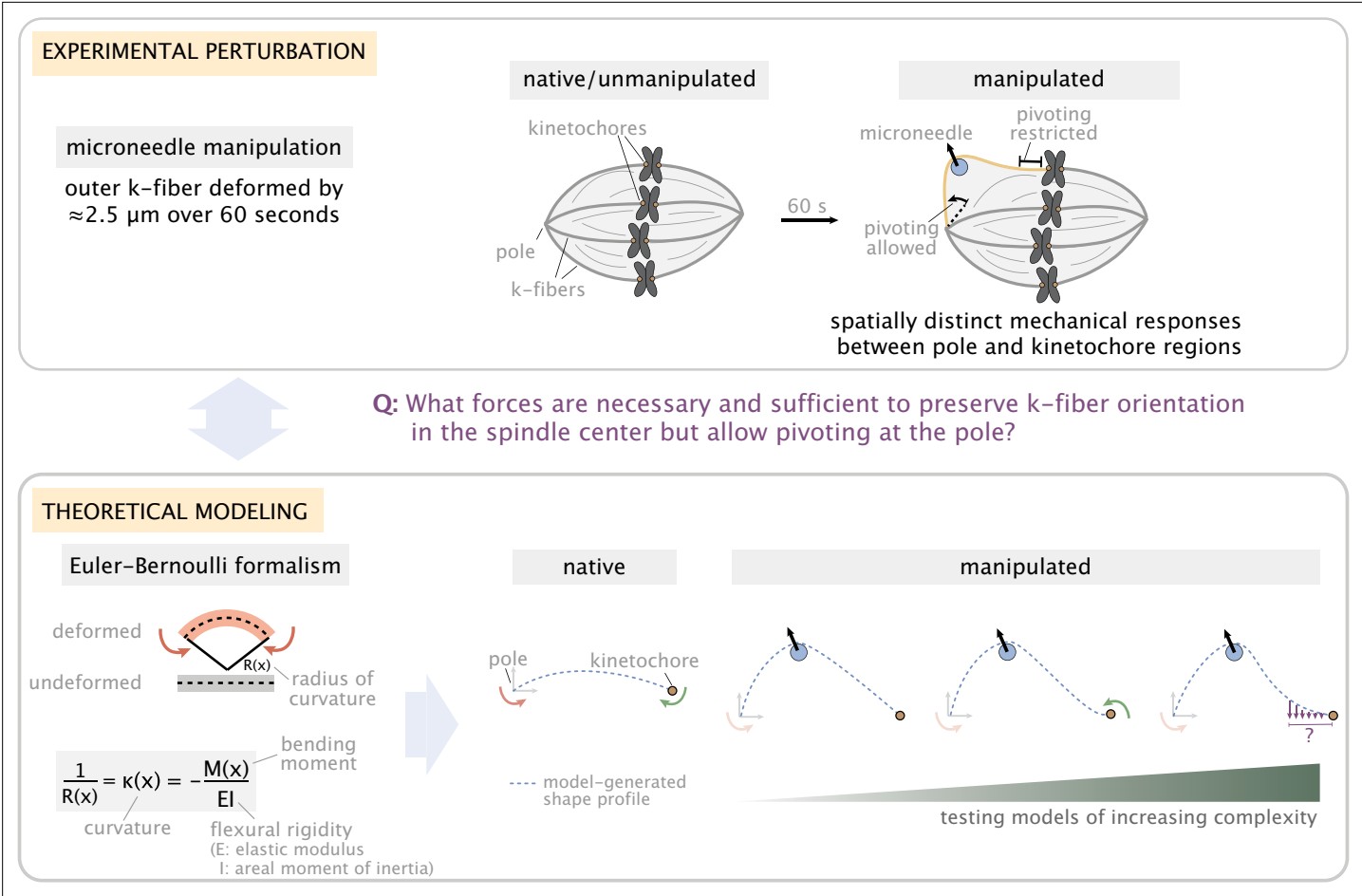

**Figure 1.** Overview of the experiment-theory interplay used for studying the mechanics of k-fiber anchorage in the mammalian spindle. Top: Schematic of the experimental perturbation performed in *Suresh et al., 2020*. Microneedle (blue circle) manipulation of outer k-fibers revealed that k-fibers do not freely pivot near kinetochores, ensuring the maintenance of k-fiber orientation in the spindle center, and pivot more freely around poles. Bottom: Coarse-grained modeling approach of the k-fiber in the spindle context based on Euler-Bernoulli beam theory. Model complexity is progressively increased to identify the minimal set of forces necessary and sufficient to recapitulate (dashed blue lines) k-fiber shapes in the data. From left to right: we test models with different forces and moments at k-fiber ends (pole and kinetochore) to recapitulate native k-fibers, and, then test models of increasing complexity (first with x- and y- force components and just a moment at the pole, then a moment at the kinetochore and finally lateral anchorage over different length scales along the k-fiber (purple arrows)) to recapitulate manipulated k-fibers. Here, forces (represented as straight arrows) and moments (represented as curved arrows) together define the bending moment M(x) along the k-fiber, while k-fiber shape is determined via curvature $\kappa$(x).

shapes, lateral anchorage is needed to preserve k-fiber orientation in the spindle center in manipulated spindles. We then systematically test different length scales of lateral anchorage. Global anchorage leads to a loss of mechanical distinction in the pole and kinetochore regions – a prediction confirmed by manipulating spindles with globally increased anchorage. In turn, local anchorage preserves the mechanical distinction observed in control manipulations, and a length scale of 3 μm near kinetochores is necessary and sufficient to recapitulate observed manipulated shapes. This localized anchorage can preserve k-fiber orientation near kinetochores without significant k-fiber-to-network detachment for a broad range of microneedle forces. Thus, strong local anchorage enacted locally within 3 μm of kinetochores can ensure that sister k-fibers remain aligned and bioriented in the spindle center robustly, while allowing their pivoting and clustering into poles. Together, we demonstrate how spatially regulated anchorage gives rise to spatially distinct mechanics, which we propose support different functions across the spindle.

## Results

### Forces and moments acting on k-fiber ends alone can capture native mammalian k-fiber shapes

To determine the spindle forces necessary and sufficient to recapitulate k-fiber shapes, we use the Euler-Bernoulli formalism of beam deformation (*Landau and Lifshitz, 1984*). Through the equation κ(x)=M(x)/EI (*Figure 1*, bottom), this formalism relates the curvature κ(x) of the beam at a given position (x) to the local bending moment M(x) (the moment of internal stresses that arise from forces exerted) and the flexural rigidity EI (a measure of resistance to bending that depends on the elastic modulus E and the areal moment of inertia I of the beam). We treat the k-fiber as a single homogeneous beam (*Rubinstein et al., 2009*; *Kajtez et al., 2016*) that bends elastically in response to force (see Materials and methods).

In the mammalian metaphase spindle, native k-fibers appear in a variety of curved shapes which arise from the molecular force generators that maintain the spindle (*Elting et al., 2018*; *Nazockdast and Redemann, 2020*; *Tolić and Pavin, 2021*). To obtain a minimal description of native k-fiber shape generation, we considered point forces and moments acting only on the pole and kinetochore ends of the k-fiber (*Rubinstein et al., 2009*). These could arise from motor and non-motor microtubule-associated proteins that exert force on and anchor k-fiber ends, for example from dynein and NuMA at poles (*Heald et al., 1996*; *Merdes et al., 1996*), and NDC80 at the kinetochore (*DeLuca et al., 2006*). In our minimal description, we coarse-grained the kinetochore-proximal forces (a tensile force at the kinetochore (*McNeill and Berns, 1981*; *Waters et al., 1996*) and a compressive force near the kinetochore (*Rubinstein et al., 2009*; *Kajtez et al., 2016*) to an effective point force (see Materials and methods)). Using a fine-grained junction model with explicit tensile and compressive forces did not significantly change the model outcomes (*Figure 2—figure supplement 1*), thereby justifying our coarse-grained approach. We considered a coordinate system where the pole is at the origin (x=y = 0) and the kinetochore lies along the x-axis at position x=L (*Figure 2a*). In this system, a force at the pole (**F** with components $F_x$ and $F_y$), an equal and opposite force at the kinetochore (at equilibrium), and a moment at the pole ($M_p$) and at the kinetochore ($M_k$) together define the shape of the k-fiber at every position **r**(x) via the moment balance condition (**M**(x)=**$M_p$** + **r**(x)×**F**, with $M_k$=M(x=L)). The relatively small deflection of native k-fibers allowed us to solve for their shape profiles analytically and gain insight into how these forces and moments uniquely contribute to shape (see Appendix 1). We found that a purely axial force $F_x$ generates a symmetric shape profile with the peak (position where the deflection y(x) is the largest) located in the middle of the pole-kinetochore axis (*Figure 2b*, top). In the absence of an axial force $F_x$, the moment at the pole $M_p$ and corresponding force $F_y$ generate an asymmetric shape profile with the peak shifted towards the pole (*Figure 2b*, middle); conversely, the moment at the kinetochore $M_k$ and corresponding force $F_y$ generate an asymmetric shape profile with the peak shifted towards the kinetochore (*Figure 2b*, bottom). This finding is consistent with the idea that each force and moment component acting on the ends uniquely contributes to k-fiber shape.

To determine which subset of force components (*Figure 2b*) is necessary and sufficient to capture native k-fiber shapes, we imaged native k-fibers in PtK2 GFP-tubulin cells at metaphase (m=26 cells, n=83 k-fibers) and extracted the distribution of peak locations along their length (*Figure 2c*). Most peaks are located closer to the pole or in the middle of the k-fiber (*Figure 2d*), suggesting that the moment $M_k$ is not essential for their shape generation. We then fit different combinations of force components in our model to the shape profiles extracted from the data (see Materials and methods). We evaluated the quality of model fits based on two metrics: fitting error (measured by calculating the root mean square error, *Figure 2e*) and comparison of peak locations between the model fit and data shape profiles (*Figure 2f*). The combination of $F_x$, $F_y$ and $M_p$ together produced the lowest fitting error (*Figure 2e*), and accurately predicted the peak locations (*Figure 2f*, example fits in *Figure 2g*), while the other subsets of force components performed significantly worse on both metrics. The inclusion of $M_k$ along with $F_x$, $F_y$ and $M_p$ did not significantly improve the quality of fits (*Figure 2—figure supplement 2*), revealing that $M_k$ is indeed not necessary to recapitulate native k-fiber shapes. Taken together, while native k-fiber shapes are diverse, the consistent shift in peaks toward the pole reveals a key mechanical role for the moment at the pole. This indicates that forces at the k-fiber ends and a moment at the pole (**F**, $M_p$), but not at the kinetochore ($M_k$ = 0), are alone necessary and sufficient to recapitulate native k-fiber shapes.

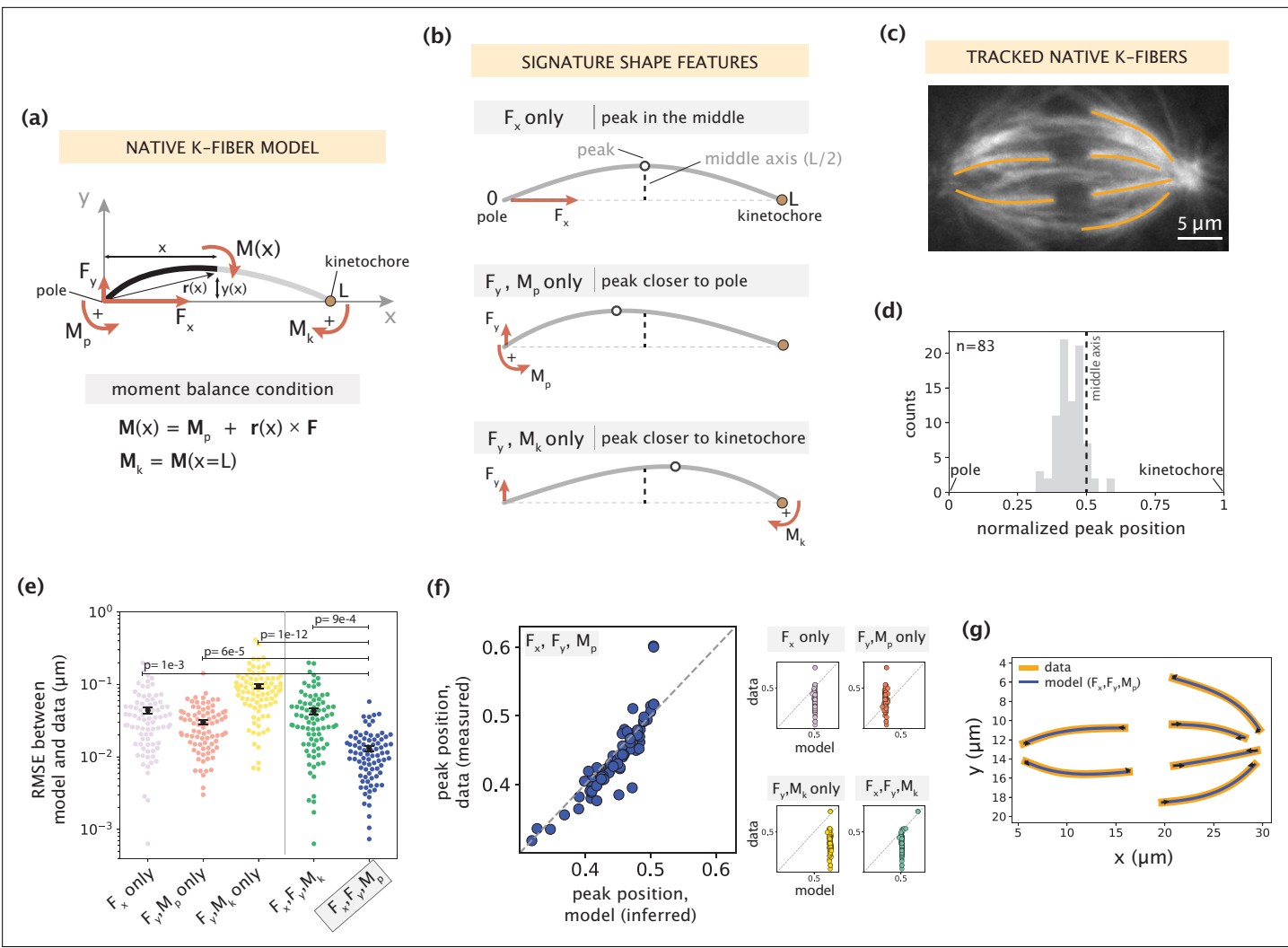

**Figure 2.** Forces and moments acting on k-fiber ends alone can capture native mammalian k-fiber shapes. See also *Figure 2—figure supplements 1–3*. (**a**) Schematic of the minimal model for native/unmanipulated k-fibers. Pole and kinetochore ends are oriented along the x-axis (x from 0 to L). Only forces ($F_x$, $F_y$; red linear arrows) and moments ($M_p$, $M_k$; red curved arrows) acting on k-fiber ends are considered. The moment balance condition M(x) shown below defines the k-fiber shape at every position via the Euler-Bernoulli equation. (**b**) The unique mechanical contribution of each model component to a signature shape feature of native k-fibers. The white circle denotes the k-fiber's peak position (location where the deflection y(x) is the largest). Each component uniquely shifts the peak position relative to the middle axis (dashed line at x=L/2). (**c**) Representative image of a PtK2 GFP-tubulin metaphase spindle (GFP-tubulin, white) with tracked k-fiber profiles overlaid (orange). (**d**) Distribution of peak positions of native k-fibers tracked from PtK2 GFP-tubulin cells at metaphase (m=26 cells, n=83 k-fibers), normalized by the k-fiber's end-to-end distance, with the middle axis (black dashed line) at x=0.5. (**e**) Root-mean-squared error (RMSE) between the experimental data (m=26 cells, n=83 k-fibers) and the model-fitted shape profiles. Plot shows mean ± SEM. (**f**) Comparison of normalized peak positions between the experimental data (m=26 cells, n=83 k-fibers) and model-fitted shape profiles for each model scenario. The model with $F_x$, $F_y$ and $M_p$ (blue points) best captures the peak positions in the data (Pearson $R^2$ coefficient = 0.85, p=7e-22). Black dashed line corresponds to an exact match of peak positions between the model prediction and measurement data. (**g**) Tracked k-fiber profiles from the spindle image (**c**) and their corresponding model fits performed with the minimal model with $F_x$, $F_y$ and $M_p$, but not $M_k$. Black arrows represent the model-inferred forces at end-points.

The online version of this article includes the following figure supplement(s) for figure 2:

**Figure supplement 1.** Coarse-graining of kinetochore-proximal tensile and compressive forces to an effective point force at the kinetochore does not alter model outcomes.

**Figure supplement 2.** The minimal model does not require a moment at the kinetochore to capture native k-fiber shapes.

**Figure supplement 3.** No detectable trend between inner and outer k-fibers is observed in the parameters inferred by the minimal model.

Having established our minimal native k-fiber model, we used it to examine how shape and force generation (**F**, $M_p$) vary across k-fiber angles with respect to the spindle's pole-pole axis (*Figure 2—figure supplement 3a*). We hypothesized that outer k-fibers, with larger angles from the pole-pole axis and which visually appear more bent, would be exposed to larger forces and moments. While k-fibers with larger angles indeed have larger deflections on average (*Figure 2—figure supplement 3b-c*), we observed no detectable trend in inferred force parameters (*Figure 2—figure supplement 3d-e*), suggesting a lack of distinction in the force generation across different k-fiber angles in the spindle. Instead, our model suggests that the greater average length of outer k-fibers (*Figure 2—figure supplement 3f*) is sufficient to capture their larger deflections (*Figure 2—figure supplement 3g*, Appendix 1.4). Thus, k-fiber length can serve as another contributor to the observed shape diversity. Together, by connecting shape to forces, we determine that point forces on k-fiber ends and a moment at the pole are sufficient to recapitulate the diverse array of native k-fibers, and postulate that force generation is not differentially regulated across k-fiber angles in the mammalian spindle.

## Manipulated k-fiber response cannot be captured solely by end-point anchoring forces and moments

Having defined a minimal model for native k-fiber shape generation, we turned to manipulated k-fibers, under the premise that mechanical perturbations can more discriminately expose underlying mechanics. We sought to determine the spindle forces necessary and sufficient to restrict the k-fiber's free pivoting near the kinetochore (reflected by a negative curvature in that region) but not near the pole when under external force (*Figure 3a*, *Suresh et al., 2020*). We included in our model an external microneedle force ($F_{ext}$) treated as a point force (Appendix 2) whose contribution to the bending moment at $r(x)$ is $(r(x) - r_{ext}) \times F_{ext}$. To build up model complexity systematically, we first tested whether the minimal spindle forces acting solely on k-fiber ends (**F**, $M_p$ with $M_k = 0$) together with $F_{ext}$ (*Figure 3b*) can capture manipulated k-fibers. We extracted the k-fiber shape profiles from GFP-tubulin PtK2 metaphase spindles under manipulation (m=18 cells, n=19 k-fibers, deformed by 2.5±0.2 μm over 60.5±8.8 s, *Figure 3—video 1*, *Suresh et al., 2020*) and fit the model (see Appendix 3 and 4 for fitting details). The model failed to capture the negative curvature region near kinetochores (*Figure 3c*), giving rise to fitting errors that are 10-fold larger than the native k-fiber model (*Figure 3f*).

We then hypothesized that introducing a negative moment $M_k$ at the kinetochore to restrict free pivoting there could be sufficient to recapitulate manipulated k-fiber shapes (*Figure 3d*). Performing fits to the data revealed that the model with $M_k$ produced a negative curvature near the kinetochore (*Figure 3e*), leading to a substantial decrease in the fitting errors compared to the model where $M_k = 0$. However, the fitting errors are still not comparable to those of the native k-fiber model (*Figure 3f*). To better evaluate the model's performance, we compared several signature shape features between the data and model. While the model with $M_k$ accurately captures the positions of positive curvature maxima (*Figure 3g*), it consistently fails to capture the positions of negative curvature minima (*Figure 3h*, example in *Figure 3e*). The positions of curvature minima in the experimental data span a range of 0.5–3 μm from the kinetochore; however, they are much more localized (within 0.5 μm) in model-generated profiles (*Figure 3h*). Similarly, the model fails to capture the region over which the k-fiber's orientation angle near the kinetochore is preserved, which spans 3 μm in the data (*Figure 3i*, *Figure 3—figure supplement 1*).

Motivated by electron microscopy studies demonstrating that k-fibers have 20–30% more microtubules near kinetochores compared to poles (*McDonald et al., 1992*; *O'Toole et al., 2020*), we also tested the impact of having a non-uniform k-fiber flexural rigidity on the position of the negative curvature minimum. We found that a local increase in flexural rigidity can shift the position of negative curvature away from the kinetochore (*Figure 3—figure supplement 2a-b*). However, for the curvature minimum be up to 3 μm away from the kinetochore, the kinetochore-proximal region would need to have twice as many microtubules than the rest of the k-fiber (*Figure 3—figure supplement 2c-d*), which is inconsistent with structural studies (*McDonald et al., 1992*; *O'Toole et al., 2020*). Taken together, these findings exclude the possibility of end-localized anchoring forces and moments being the sole contributors to the response features observed in manipulated k-fibers.

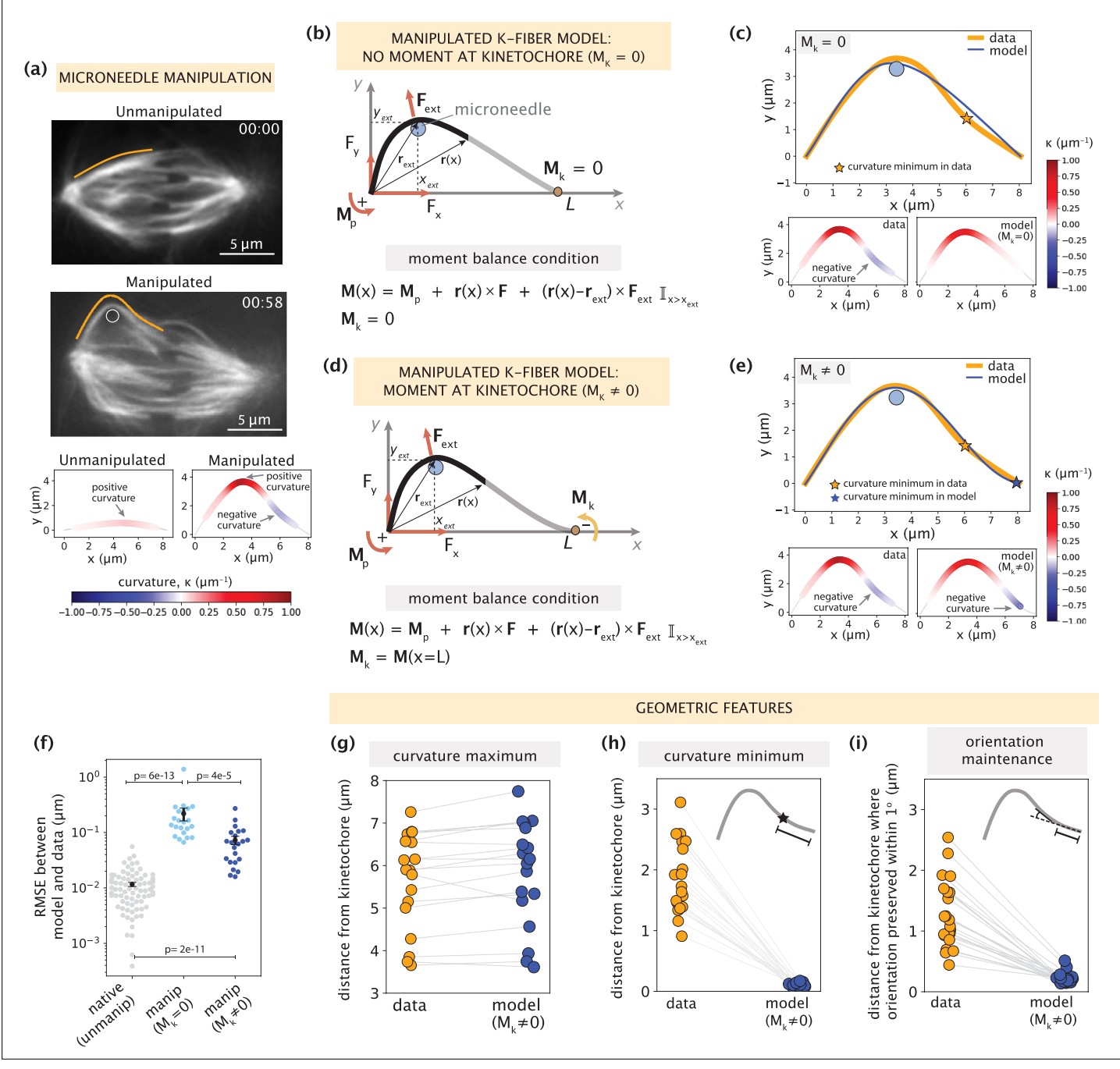

**Figure 3.** Manipulated k-fiber response cannot be captured solely by end-point anchoring forces and moments. See also *Figure 3—figure supplements 1–2* and *Figure 3—video 1*. (**a**) Top: Representative images of a PtK2 spindle (GFP-tubulin, white) before and at the end of microneedle manipulation. Tracked k-fiber profiles (orange) and the microneedle (white circle) are overlaid (shifted for k-fiber) on the images. Bottom: Curvature profiles along the native/unmanipulated and manipulated k-fibers. Time in min:sec. Scale bar = 5 μm. (**b**) Schematic of the model for the manipulated k-fiber that includes $\mathbf{F}$ ($F_x$, $F_y$), $M_p$ and $M_k$ is set to zero (minimal native k-fiber model), along with an external force $\mathbf{F}_{ext}$ from the microneedle (blue circle). The moment balance condition is shown below, where the indicator function (I) specifies the region over which the corresponding term in the equation contributes to $\mathbf{M}(x)$. (**c**) Top: Manipulated shape profile extracted from the image in (**a**) (orange line), together with the best fit profile generated by the model (blue line) where $M_k = 0$. Stars denote the minimum of the negative curvature. The model does not capture the negative curvature observed in the data (orange star). Bottom: Curvature profiles along the k-fiber in the data (left) and the model (right). (**d**) Schematic of the model for the manipulated k-fiber defined by the parameters in (**b**) and a negative moment at the kinetochore, $M_k$ (orange arrow). (**e**) Top: Manipulated shape profile extracted from the image in (**a**) (orange line), together with the best fit profile generated by the model (blue line) with $M_k \neq 0$. Stars denote the minimum of the negative curvature. The model generates a negative curvature (blue star) but cannot accurately capture its position from the data (orange

*Figure 3 continued on next page*

Figure 3 continued

star). Bottom: Curvature along the k-fiber in the data (left) and the model (right). (**f**) Root-mean-squared error (RMSE) between the experimental data (m=18 cells, n=19 k-fibers) and the best fitted profiles from the models without ($M_k = 0$) and with ($M_k{\neq}0$) a moment at the kinetochore. A comparison is made also with the RMSE of the minimal native k-fiber model (control, **Figure 2a**). Plots show mean ± SEM. (**g–i**) Comparison of manipulated k-fiber profiles (m=18 cells, n=19 k-fibers) between the data and model ($M_k{\neq}0$) for (**g**) positions curvature maxima (Pearson $R^2$ coefficient = 0.95, p=4e-13), (**h**) positions of curvature minima (Pearson $R^2$ coefficient = 0.27, p=1e-1), and (**i**) the distance over which the orientation angle is preserved within 1° (Pearson $R^2$ coefficient = 0.43, p=9e-4).

The online version of this article includes the following video and figure supplement(s) for figure 3:

**Figure supplement 1.** K-fiber orientation in model profiles with $M_k{\neq}0$ is preserved over much shorter distances than in experimental data, irrespective of the chosen threshold angle.

**Figure supplement 2.** Impact of nonuniform flexural rigidity on the k-fiber response to force.

**Figure 3—video 1.** Microneedle manipulation of PtK2 metaphase spindles reveals the restriction of k-fiber pivoting around the kinetochore but not the pole.

https://elifesciences.org/articles/79558/figures#fig3video1

## Mapping the relationship between anchorage length scales and manipulated k-fiber shapes constrains the spatial distribution of lateral anchorage

To determine the forces needed to preserve k-fiber orientation at a relevant length scale in the spindle center (**Figure 3h–i**) and to also capture the observed mechanical distinction between the kinetochore and pole regions, we investigated how lateral anchorage along the k-fiber's length influences the k-fiber's mechanical behavior. We sought to systematically vary the spatial distributions of lateral anchorage and map the k-fiber's response to force. Our previous work revealed that the crosslinking protein PRC1, which preferentially binds antiparallel microtubules and helps organize bridging-fibers (**Jagrić et al., 2021**), plays a key role in mediating the lateral anchorage responsible for negative curvature near kinetochores (**Suresh et al., 2020**). However, how the absolute levels of PRC1 along k-fibers (**Polak et al., 2017**; **Suresh et al., 2020**) map to mechanical anchorage is unknown, thus motivating the need to directly vary lateral anchorage in space.

We enhanced our model, treating the anchoring network to which the k-fiber is coupled as a uniformly distributed series of elastic springs which exert restoring forces **f**(x) along the region of anchorage (**Figure 4a**). In our treatment, the anchoring network does not detach from the k-fiber (see Materials and methods). In a simulation study, we systematically tuned the length scale of lateral anchorage near the kinetochore (σ=1–10 µm), and initially considered a step function distribution of anchorage present only within the region L-σ<x < L. Mimicking our previous experimental procedure (**Figure 4b**, **Suresh et al., 2020**), we also tuned the position of the microneedle.

K-fiber shape profiles simulated with different anchorage length scales σ revealed a broad array of negative curvature responses, where the positions of curvature minima were strongly affected by the choice of σ (**Figure 4c**). To probe the relationship between the anchorage length scale and the k-fiber's response to force, we compared these simulated k-fiber shape profiles to manipulation experiments in control spindles (**Suresh et al., 2020**) and spindles where crosslinking was globally increased experimentally. Consistent with wildtype spindle manipulations (2.5±0.2 µm over 60.5±8.7 s, **Suresh et al., 2020**), simulated shapes with local anchorage (example of σ=2 µm for 10 µm long k-fiber in **Figure 4d**) had a negative curvature response only near the kinetochore (and not near the pole) that remained localized for a range of microneedle positions, thereby generating spatially distinct mechanical responses between the pole and kinetochore regions. Local anchorage with σ=1–3 µm near the kinetochore best captures the range of experimentally observed curvature minima positions (**Figure 4c**). This conclusion holds true for a wide range of chosen anchorage strengths (**Figure 4— figure supplement 1**). On the other hand, simulated shapes with global anchorage (example of σ=10 µm along the entire k-fiber length in **Figure 4e**) had negative curvature on both the kinetochore and pole sides of the microneedle, leading to a loss of mechanical distinction between these two regions. Additionally, with global anchorage the curvature minima positions do not remain localized near the kinetochore but rather move with the microneedle position. To test this experimentally, we globally increased crosslinking with FCPT treatment – a drug that rigor-binds kinesin-5 to microtubules (**Groen et al., 2008**). Consistent with the global anchorage model predictions, manipulations

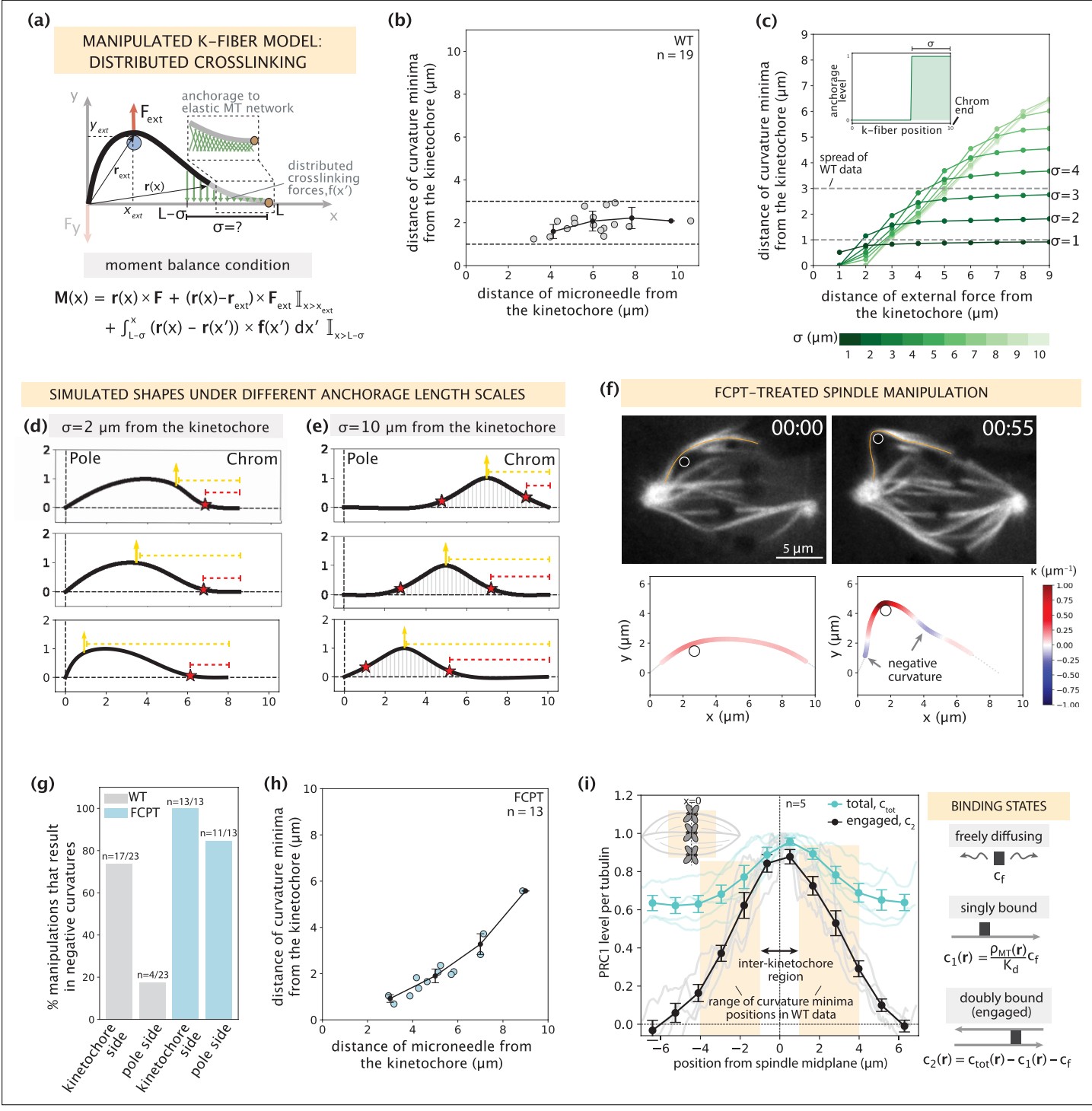

**Figure 4.** Mapping the relationship between anchorage length scales and manipulated k-fiber shapes constrains the spatial distribution of lateral anchorage. See also *Figure 4—figure supplements 1–3* and *Figure 4—video 1*. (**a**) Schematic of the model for the manipulated k-fiber with crosslinking forces f(x') distributed over a length scale σ near the kinetochore. The model also includes endpoint forces F, and an external force from the microneedle $F_{ext}$. The crosslinking force density is f(x') = -k y(x') ŷ, where k is the effective spring constant and ŷ is the unit vector in the y direction. Since we do not expect $M_p$ to influence the crosslinking behavior near the kinetochore, for simplicity we set $M_p = 0$ in simulation studies of this section. Indicator function (I) in the moment balance condition specifies the region over which the corresponding term contributes to M(x). (**b**) Distance of curvature minima as a function of distance of the microneedle from the kinetochore (m=18 cells, n=19 k-fibers), in wildtype spindle manipulations (*Suresh et al., 2020*). Plot shows mean ± SEM (black). (**c**) Distance of curvature minima as a function of distance of external force application from the kinetochore calculated for model-simulated profiles where the length scale of anchorage (σ, inset) is tuned in the range 1–10 μm (denoted by shades

*Figure 4 continued on next page*

*Figure 4 continued*

of green). Variation of the position of external force application mimics the wildtype manipulation experiments in (**b**). Dashed lines denote the spread of curvature minima positions in (**b**). (**d,e**) Profiles generated by the model in (**a**) with (**d**) σ=2 μm and (**e**) σ=10 μm for varying positions of external force application (yellow arrow), and the resulting positions of curvature minima (red star). Dashed lines represent the distances from the kinetochore to the external force position (yellow dashed line) and curvature minimum position (red dashed line). (**f**) Top: Representative images of a PtK2 spindle (GFP-tubulin, white) treated with FCPT to rigor-bind the motor Eg5, in its unmanipulated (00:00) and manipulated (00:55) states. The microneedle (white circle) and tracked k-fibers (orange) are displayed on images. Bottom: Curvature along the tracked k-fibers. Time in min:sec. (**g**) Percentage of microneedle manipulations that gave rise to a negative curvature near the kinetochore and the pole in wildtype (grey; m=18 cells, n=19 k-fibers, *Suresh et al., 2020*) and FCPT-treated (light blue; m=11 cells, n=13 k-fibers) spindles. (**h**) Distance of curvature minima as a function of distance of the microneedle from the kinetochore in FCPT-treated spindle manipulations (m=9 cells, n=13 k-fibers). Plot shows mean ± SEM (black). (**i**) Left: Normalized distribution of PRC1's total abundance levels ($c_{tot}$, cyan lines) measured from immunofluorescence images (fluorescence intensity, n=5 cells) (*Suresh et al., 2020*) and actively engaged (doubly bound, $c_2$, grey lines) PRC1 calculated from the $c_{tot}$ using the equilibrium binding model along the spindle's pole-pole axis (x=0 represents the spindle midplane). The region along k-fibers where negative curvature is observed in the wildtype dataset is highlighted in orange and the inter-kinetochore region (double-sided black arrow) denotes the chromosome region between the sister k-fibers (inset). Plot shows mean ± SEM for both PRC1 populations. Right: Three distinct binding states of PRC1 considered in our analysis. The concentration of actively engaged PRC1 ($c_2$) is calculated by subtracting the free and singly bound contributions from the total PRC1 concentration ($c_{tot}$). In the expression for the singly-bound PRC1 population, $\rho_{MT}(r)$ stands for the local tubulin concentration, while $K_d$ represents the dissociation constant of PRC1– single microtubule binding.

The online version of this article includes the following video and figure supplement(s) for figure 4:

**Figure supplement 1.** Tuning anchorage strength within an order of magnitude only introduced sub-micron variations in the negative curvature response.

**Figure supplement 2.** Microneedle displacement over time in FCPT-treated spindle manipulations.

**Figure supplement 3.** Negative curvature does not remain localized near chromosomes when anchorage levels in the model mimic PRC1 intensity in the spindle.

**Figure 4—video 1.** Microneedle manipulation of FCPT-treated PtK2 spindles reveals negative curvature on both sides of the microneedle and not localized near kinetochores.

https://elifesciences.org/articles/79558/figures#fig4video1

in FCPT-treated spindles (2.7±0.1 μm over 59.6±2.7 s in GFP-tubulin PtK2 cells, m=10 cells, n=13 k-fibers) (*Figure 4f*, *Figure 4—figure supplement 2*) resulted in negative curvature on both sides of the microneedle (*Figure 4g*), and its position moved as the microneedle was moved (*Figure 4h*, *Figure 4—video 1*). Thus, local anchorage is required to capture both the spatially distinct mechanics and localized nature of the negative curvature response observed in wildtype manipulated k-fibers.

Given this finding, and PRC1's known role in localized anchorage (*Suresh et al., 2020*), we asked if an anchorage distribution reflecting PRC1's abundance in the spindle is sufficient to capture the localized negative curvature response. Mimicking PRC1 levels from immunofluorescence imaging (*Suresh et al., 2020*), we set the length scale of enrichment to be σ=3 μm from the kinetochore, and the basal anchorage elsewhere to be 60% of this enriched region. Assuming PRC1 molecules are equally engaged everywhere, our model predicted that the curvature minimum moves with the microneedle (*Figure 4—figure supplement 3*), contrary to our experimental observation (*Figure 4b*). Together with the finding that PRC1 is required for the manipulation to result in a negative curvature response (*Suresh et al., 2020*), this suggests that PRC1's crosslinking engagement varies over space, and that its abundance is not a good proxy for its mechanical engagement.

To probe how local or global the mechanical engagement of PRC1 is in the spindle and gain intuition on how this gives rise to the observed localized negative curvature response, we proceeded to more precisely define the region over which PRC1 actively crosslinks microtubules. While the precise spatial distribution of PRC1 engagement cannot be directly measured in vivo, we sought to extract this information from immunofluorescence data (*Suresh et al., 2020*) using an equilibrium binding model. Specifically, we distinguished between the doubly bound ($c_2(\mathbf{r})$, actively crosslinking two microtubules), singly bound ($c_1(\mathbf{r})$, on one microtubule but not crosslinking), and freely diffusing ($c_f$) states from the measured total ($c_{tot}(\mathbf{r})$) PRC1 abundance (Appendix 5). Based on the facts that PRC1 binds much more weakly to parallel microtubules (30-fold lower affinity than to antiparallel microtubules *Bieling et al., 2010*), and that microtubules near poles are predominantly parallel (*Euteneuer and McIntosh, 1981*), we considered PRC1 engagement in this region to be negligible. Under these conditions, the model infers the actively engaged PRC1 ($c_2(\mathbf{r})$) to be predominantly in the spindle center and substantially lower away from the center (*Figure 4i*). This is akin to the local anchorage

scenarios without basal levels (tested in *Figure 4c and d*) and suggests that while PRC1's enrichment on top of a basal level cannot give rise to a localized negative curvature response (*Figure 4—figure supplement 3*), its locally distributed mechanical engagement can do so.

Taken together, systematically exploring the k-fiber responses that arise from different anchorage length scales revealed the need for lateral anchorage to be local, and defining PRC1's abundance-to-anchorage relationship helped demonstrate how it could provide such local anchorage.

## Minimal k-fiber model infers strong lateral anchorage within 3 µm of kinetochores to be necessary and sufficient to recapitulate manipulated shape profiles

Having demonstrated the essential role of local anchorage in producing a negative curvature k-fiber response near the kinetochore, we investigated if its inclusion in our minimal k-fiber model is sufficient to recapitulate all response features of manipulated k-fibers; and if so, over what length scale does this anchorage need to be? Because of the challenges in extracting an accurate strain map of the anchoring network and knowledge of the baseline state before and after manipulation, using a model with distributed springs (*Figure 4a*) that would require these as input information, was not feasible. Further, a simple one-to-one mapping between the undeformed and deformed k-fibers could not be done due to the k-fiber length changing over the 60 s manipulation. We therefore captured the collective influence of localized anchorage forces using an effective point crosslinking force $\mathbf{F}_c$ (*Figure 5a*). This approach allows us to learn about both the mechanics and spatial regulation of anchorage, while being agnostic of the network's constitutive law and also simplifying the parameter search. We validated this coarse-grained approach by simulating k-fiber shapes with different local anchorage distributions near the kinetochore, and performing fits to these shapes using the minimal model (*Figure 3b*) with now $\mathbf{F}_c$ in place (*Figure 5—figure supplement 1a*). Indeed, the fits revealed the inferred magnitude of $\mathbf{F}_c$ to be close to the integrated anchorage force (*Figure 5—figure supplement 1b*), and its position $x_c$ (a distance $\lambda_c$ from the kinetochore) to be consistently proximal to the edge of the localized anchorage region σ where anchorage forces are the largest (*Figure 5—figure supplement 1c-d*). Thus, an effective point crosslinking force $\mathbf{F}_c$ can successfully coarse-grain locally distributed anchorage forces.

We then fit the model with an effective point force $\mathbf{F}_c$ to all observed manipulated k-fiber shape profiles. In all but four cases with significantly large positive curvature values (*Figure 5—figure supplement 2*), which could be suggestive of local fracturing due to the microneedle force (*Schaedel et al., 2015*), the model accurately recapitulated the data (*Figure 5b*). This is reflected in the significantly lower fitting errors compared to the previous manipulated k-fiber models (*Figure 5c*). To better evaluate the model's performance, we compared several signature shape features between the data and model. The curvature maxima and minima positions (*Figure 5d–e*, example in *Figure 5b* right), and length scale over which k-fiber orientation is preserved were all captured accurately (*Figure 5f*). Thus, an effective point crosslinking force ($\mathbf{F}_c$) that coarse-grains the local anchorage near the kinetochore, together with $\mathbf{F}$, $M_p$ and $\mathbf{F}_{ext}$, define the minimal model sufficient to recapitulate the shapes of manipulated k-fibers.

Next, we investigated the length scale of lateral anchorage inferred by the minimal model to recapitulate manipulated k-fiber shapes. Across all manipulated k-fibers in the dataset, the model infers $\lambda_c$ (which directly informs on the anchorage length scale *Figure 5—figure supplement 1d*) to be consistently within 3 µm of kinetochores (*Figure 5g*), indicating that this length scale of lateral anchorage is necessary and sufficient to robustly restrict k-fiber pivoting across the spindle center without obstructing pivoting at poles. This result is in close agreement with the anchorage length scales predicted from the simulated shapes (*Figure 4c*) and the region where actively engaged PRC1 is predicted to be predominantly present (*Figure 4i*). We also identified a strong correlation between the inferred anchorage length scale ($\lambda_c$) and curvature minimum position (*Figure 5h*). While previously we associated the occurrence of negative curvature with the presence of anchorage (*Suresh et al., 2020*), this finding now offers an interpretation for the position of curvature minimum as a quantitative predictor of the length scale of local anchorage.

Finally, having arrived at a minimal model sufficient to recapitulate the k-fiber's response to manipulation, we investigated how the anchoring network responds to microneedle forces using the results of model inference. Our model inference revealed that in response to microneedle forces ranging

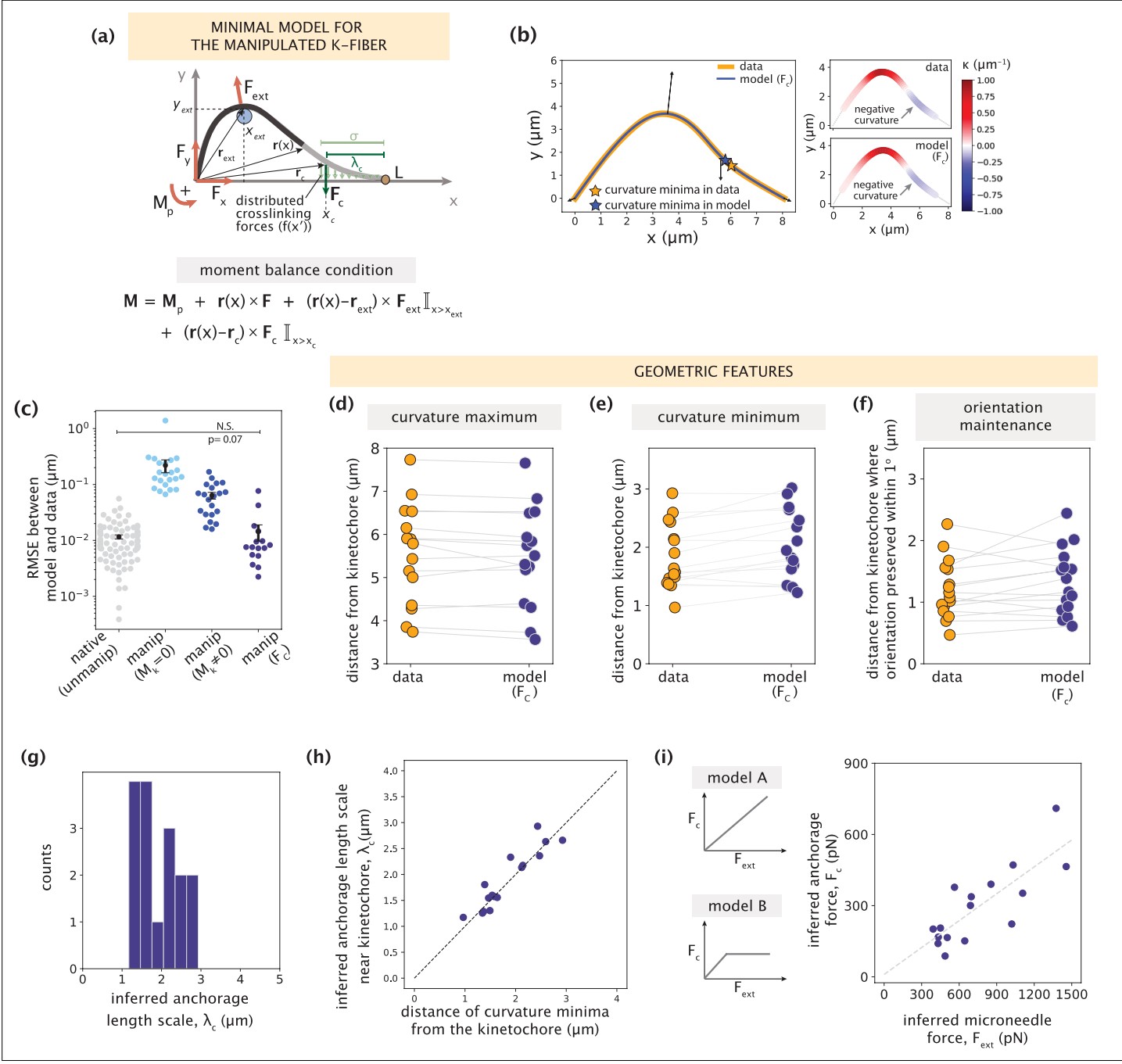

**Figure 5.** Minimal k-fiber model infers strong lateral anchorage within 3 μm of kinetochores to be necessary and sufficient to recapitulate manipulated shape profiles. See also *Figure 5—figure supplements 1–2*. (**a**) Schematic of the model for the manipulated k-fiber with an effective point crosslinking force ($F_c$, dark green arrow) a distance $\lambda_c$ away from the kinetochore-end introduced to capture the effect of the distributed crosslinking forces (light green arrows) localized near the kinetochore. The model also includes the parameters F, $M_p$ and $F_{ext}$. Indicator function (I) in the moment balance condition specifies the region over which the corresponding term contributes to M(x). (**b**) Left: Manipulated shape profile extracted from the image in *Figure 3a* (orange line), overlaid with the best fitted profile inferred by the model with $F_c$ (blue line). Black arrows represent the model-inferred forces that correspond to (**a**). Stars denote the minimum of the negative curvature, which matches well between the data (orange star) and model (blue star). Right: Curvature along the k-fiber in the data (top) and the model (bottom). (**c**) Root-mean-square error (RMSE) between the experimental data and all k-fiber models tested: $M_k = 0$ (*Figure 3b*), $M_k \neq 0$ (*Figure 3d*) and $F_c$ (*Figure 5a*). A comparison is made with the minimal native k-fiber model (control, *Figure 2a*). Plot shows mean ± SEM. (**d-f**) Comparison of manipulated k-fiber profiles (m=14 cells, n=15 k-fibers) between the data and model ($F_c$) for (**d**) positions curvature maxima (Pearson $R^2$ coefficient = 0.97, p=8e-12), (**e**) positions of curvature minima (Pearson $R^2$ coefficient = 0.9, p=1e-7), and (**f**) the distance over which the orientation angle is preserved within 1° (Pearson $R^2$ coefficient = 0.85, p=2e-4). Grey lines link the corresponding profiles.

*Figure 5 continued on next page*

*Figure 5 continued*

(**g**) Distribution of the length scales of anchorage ($\lambda_c$) inferred by the minimal model with $F_c$ for all k-fibers in the data. (**h**) Positions of curvature minima extracted from data profiles vs. the location of the effective crosslinking force near kinetochores inferred by the model (Pearson $R^2$ coefficient = 0.85, p=1e-6), with the black dashed line representing perfect correspondence between them. (**i**) Left: Possible scenarios for models of how anchorage force $F_c$ might correlate with the microneedle force $F_{ext}$ – linearly as is characteristic to an elastic response (model A), or linearly up to a force threshold, beyond which detachment of anchorage occurs (model B). Right: Microneedle force $F_{ext}$ vs. anchorage force $F_c$ inferred from the model shows a monotonic relationship. Grey dashed line represents the best-fit line (Spearman R coefficient = 0.85, p=4e-4).

The online version of this article includes the following figure supplement(s) for figure 5:

**Figure supplement 1.** Validating the use of a point crosslinking force to capture distributed anchorage.

**Figure supplement 2.** Minimal model with point crosslinking force fails to recapitulate the data only in cases with large positive curvature values.

from 400 pN to 1500 pN (that cause k-fiber deformations ($y_{max}$) up to ≈ 5 times native deformations), the anchoring network generated forces ranging from 100 pN to 700 pN to resist pivoting in the spindle center (see Materials and methods). Interestingly, we found a linear relationship between the inferred crosslinking force ($F_c$) and microneedle force ($F_{ext}$). This linear dependence does not plateau beyond a certain microneedle force, which would have been indicative of detachment from the k-fiber (*Figure 5i*, consistent with model A but not model B). This indicates that under the assumptions of our model, the anchoring network is strong enough to withstand large microneedle forces (producing $y_{max}^{manip}/y_{max}^{native} \gg 1$) without significant detachment from the k-fiber. Parameter inference from our minimal model therefore provides physical intuition for how the anchoring network can restrict k-fiber pivoting near kinetochores.

Altogether, by systematically building up complexity to determine the minimal model that can recapitulate k-fiber shapes under manipulation, our work sheds light on the spatial regulation and mechanics of anchorage necessary and sufficient for robust k-fiber reinforcement in the spindle center.

## Discussion

The k-fiber's ability to be dynamic and generate and respond to forces while robustly maintaining its connections and orientation within the spindle is critical for accurate chromosome segregation. Here, we asked (*Figure 1*): where along the k-fiber are its connections necessary and sufficient to robustly preserve its orientation in the spindle center while allowing pivoting at poles? We determined that while end-forces and moments can recapitulate unmanipulated k-fibers (*Figure 2*), they are insufficient to capture the manipulated k-fiber's response. Specifically, without lateral anchorage, the model fails to robustly restrict the k-fiber's pivoting throughout the spindle center region (*Figure 3*). In turn, having anchorage all along the k-fiber's length restricts pivoting at poles (*Figure 4*). Thus, in both cases, the signature mechanical distinction between the pole and kinetochore regions is lost. Our minimal model revealed that local anchorage within 3 μm of kinetochores is necessary and sufficient to accurately recapitulate the spatially distinct response of manipulated k-fibers, and that this length scale can be quantitatively inferred from the location of negative curvature, a signature shape feature of anchorage (*Figure 5*). Such reinforcement near kinetochores is well suited to ensure that sister k-fibers remain aligned with each other and bi-oriented in the spindle center, and can at the same time pivot and cluster into poles. Thus, by combining theory based on shape analysis and perturbations that expose underlying mechanics, our work provides a framework to dissect how spindle architecture gives rise to its robust and spatially distinct mechanics (*Figure 6*), and ultimately function.

The minimal model for native k-fibers enabled us to explore the physical mechanisms underlying force generation and k-fiber shapes within the spindle (*Figure 2*). It provides a framework to connect molecular-scale and cellular-scale spindle mechanics and better understand the origins of **F** and $M_p$ and of shape diversity across k-fibers and spindles. For example, it has been long known that NuMA and dynein focus microtubules at poles (*Heald et al., 1996*; *Merdes et al., 1996*); indeed, perturbing these proteins leads to straighter k-fibers (*Wittmann and Hyman, 1998*; *Howell et al., 2001*; *Elting et al., 2017*; *Guild et al., 2017*) and altered spindle shapes (*Oriola et al., 2020*). How these molecules individually and together dictate native k-fiber shapes in mammalian spindles, and what their role is in the moment generation inferred at the pole $M_p$, are exciting questions for future work. In addition to molecular forces playing a role in k-fiber shape generation and diversity, our study proposes that diverse k-fiber lengths from their arrangement within the spindle (inner vs. outer) can lead to diverse

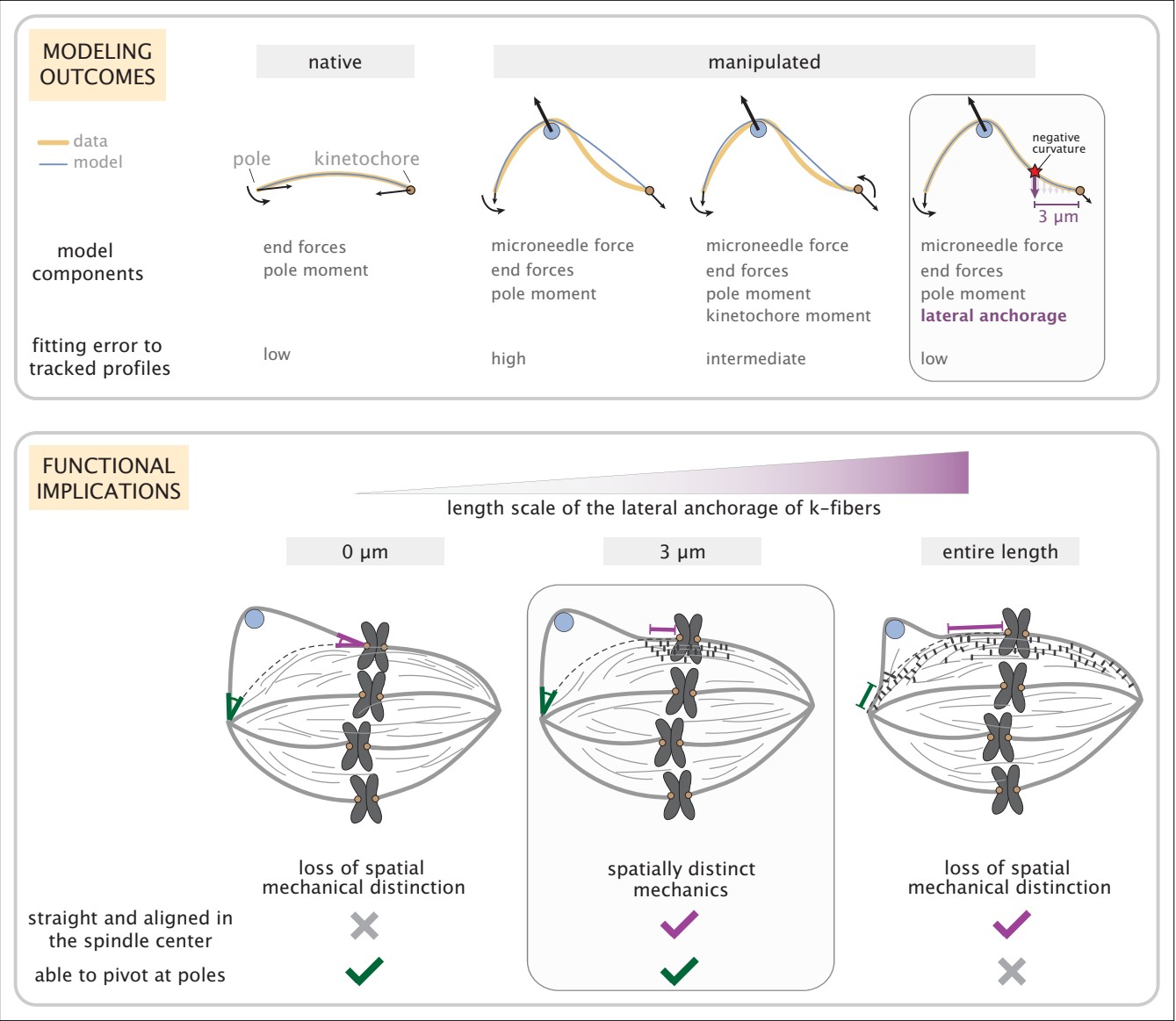

**Figure 6.** Coarse-grained modeling of k-fiber shapes reveals how spatially regulated k-fiber anchorage gives rise to spatially distinct mechanics across the mammalian spindle. Top: A summary of outcomes for the native k-fiber model and the various iterations of the manipulated k-fiber model, where we systematically built up model complexity (left to right) to capture the observed shapes. The minimal model (rightmost panel), which produced the best fits, includes forces at k-fiber ends, a moment at the pole, and localized crosslinking forces (captured through an effective point crosslinking force). The minimal model (rightmost panel) also revealed a quantitative and predictive link between the position of the negative curvature (red star) and the length scale of k-fiber anchorage (position of purple arrow from the kinetochore-end). Bottom: Functional implications of models with different length scales of anchorage (0, 3, 10 μm from left to right) tested in our study. Unlike the scenarios with no anchorage and anchorage along the entire length (left and right panel), anchorage up to 3 μm from the kinetochore (middle panel) is best suited to ensure that k-fibers remain straight in the spindle center and aligned with their sister (purple line), while also allowing them to pivot and focus at the pole (green pivot point).

k-fiber shapes. This motivates better understanding the role of other architectural features that vary across species (e.g. presence or absence of poles, spindle size, chromosome number) in contributing to k-fiber shape (*Helmke et al., 2013*; *Crowder et al., 2015*). Addressing these questions will shed light on the mechanisms ensuring robust spindle structure and function across evolution.

Our work focused on lateral anchorage in space, and revealed that local anchorage within 3 μm of kinetochores ensures that sister k-fibers remain straight in the spindle center (*Figure 5*). This could, for example, promote biorientation of chromosomes, and ultimately their accurate segregation. Further, the presence of lateral anchorage across the center of the spindle can constrain chromosome oscillations, while still allowing movement on a longer timescale and ensuring sister k-fiber

alignment. This together offers a potential explanation for why anchorage of this precise length scale can provide a robust connection to the dynamic spindle, and raises the question of how this length scale varies across spindles with different metaphase chromosome movement amplitudes. Additionally, as the dynamic k-fiber plus-end is constantly growing and shrinking (*Saxton et al., 1984*), connections between the k-fiber and anchoring microtubule network naturally break. At least some of these connections also turnover rapidly, on a seconds timescale (*Subramanian et al., 2010*; *Pamula et al., 2019*), compared to the minutes timescale of chromosome movement. Thus, having an array of connections spanning 3 μm (rather than a very localized length scale or having no lateral anchorage (*Figure 3*)) can ensure that at least some of them are still present and engaged to robustly reinforce the spindle center. In turn, not having similarly strong lateral anchorage in the pole region (*Figure 4*) can allow k-fibers and other microtubules to flexibly pivot and cluster effectively at the poles, which is thought to be important for spindle structural maintenance, and bring chromosomes to daughter cells. Taken together, spatially regulated lateral anchorage is well suited to enable different functions across different regions in the spindle (*Figure 6*).

In addition to the spatial regulation of anchorage, mechanical properties of the anchoring network are also critical for our understanding of how k-fibers respond to force. Our model revealed a linear relationship between inferred microneedle forces and anchorage force from the network in the regime probed, characteristic of an elastic response (*Figure 5i*). While individual crosslinker detachment (*Forth et al., 2014*; *Pyrpassopoulos et al., 2020*) in the network must occur, such behavior does not dominate the collective response to microneedle force. How the architecture of the non-kMT network and biophysical properties (ability to withstand and respond to force *Yusko and Asbury, 2014*) of the many motor and non-motor proteins within it dictate the network's heterogeneous mechanics (*Shimamoto et al., 2011*; *Takagi et al., 2019*) is an open question. Answering these questions for the mammalian spindle will require probing the physical (*Belmonte et al., 2017*; *Oriola et al., 2018*) and molecular (*Kajtez et al., 2016*; *Elting et al., 2017*; *Suresh et al., 2020*; *Risteski et al., 2021*) basis of the anchoring network's emergent properties, to which controlled mechanical (such as microneedle manipulation) and molecular perturbations (*Jagrić et al., 2021*) as well as modeling approaches (*Nedelec and Foethke, 2007*) will be key. Looking forward, experiments and modeling will also be useful in shedding light on the temporal dynamics of anchorage mechanics – for example, how the timescale of network relaxation relates to the kinetics of molecular turnover (*Saxton et al., 1984*; *Pamula et al., 2019*) and the manipulation protocol.

Finally, we developed our model under a set of assumptions, and relaxing some of them will provide new opportunities to test the role of additional features in determining k-fiber shape. First, we assumed that the k-fiber is mechanically homogeneous along its length. Electron microscopy of spindles revealed that k-fiber microtubules decrease in number closer to the pole (*McDonald et al., 1992*), and that their length and organization can vary depending on the system (*O'Toole et al., 2020*, *Kiewisz et al., 2021*). These factors can affect the k-fiber's flexural rigidity along its length (*Ward et al., 2014*). Second, we assumed that the k-fiber bends elastically in response to microneedle force. Forces from the microneedle could create local fractures in the microtubule lattice that leads to softening at the site of force application (*Schaedel et al., 2015*), and indeed, performing manipulations with larger deformations over longer timescales result in complete breakage of the k-fiber (*Long et al., 2020*). Exploring the contributions of a spatially variable flexural rigidity due to changes in microtubule number or local softening will help our understanding of how k-fiber mechanics and remodeling affect its response to force. Third, we only consider forces and moments that influence k-fiber shape in two dimensions. Looking forward, it will be useful to expand our model to include the potential effects of torsional forces on k-fiber shape generation (*Novak et al., 2018*). More broadly, the ability to measure forces with force-calibrated microneedles (*Nicklas, 1983*; *Shimamoto et al., 2011*) in mammalian spindles will not only help test some of these assumptions but also further refine our modeling framework.

Based on our work, we propose spatial regulation of anchorage as a simple principle for how the spindle can provide differential reinforcement across its regions to support spatially distinct core functions needed to maintain its mechanical integrity. More broadly, our work demonstrates the combination of mechanical perturbation experiments and coarse-grained modeling as a useful strategy for uncovering the mechanical design principles underlying complex cellular systems.

## Materials and methods
### Data collection and acquisition

Most of the experimental observations that motivate this work are from *Suresh et al., 2020*. The new experiments performed in this work (*Figure 4f–h*) were performed consistently with these experiments.

### Cell culture

Experiments were performed using PtK2 GFP-α-tubulin cells (stable line expressing human α-tubulin in pEGFP-C1, Clontech Laboratories, Inc; a gift from A Khodjakov, Wadsworth Center, Albany, NY *Khodjakov et al., 2003*), which were cultured as previously reported (*Suresh et al., 2020*). The cell line tested negative for mycoplasma.

### Drug/dye treatment

For the study in *Figure 4f–h* where we investigated the k-fiber's response to force under increased global crosslinking, we treated cells with FCPT (2-(1-(4-fluorophenyl)cyclopropyl)–4-(pyridin-4-yl) thiazole) (gift of T Mitchison, Harvard Medical School, Boston, MA), which rigor binds Eg5 (*Groen et al., 2008*). Cells were incubated with 20 µM of FCPT for 15–30 min before imaging.

### Imaging

PtK2 GFP-α-tubulin cells were plated on 35 mm #1.5 coverslip glass-bottom dishes coated with poly-D-lysine (MatTek, Ashland, MA) and imaged in $CO_2$-independent MEM (Thermo Fisher). Cells were maintained at 27–32°C in a stage top incubator (Tokai Hit, Fujinomiyashi, Japan), without a lid. Live imaging was performed on a CSU-X1 spinning-disk confocal (Yokogawa, Tokyo, Japan) Eclipse Ti-E inverted microscopes (Nikon) with a perfect focus system (Nikon, Tokyo, Japan), and included the following components: head dichroic Semrock Di01-T405/488/561, 488 nm (150 mW) and 561 (100 mW) diode lasers (for tubulin and microneedle respectively), emission filters ETGFP/mCherry dual bandpass 59022 M (Chroma Technology, Bellows Falls, VT), and Zyla 4.2 sCMOS camera (Andor Technology, Belfast, United Kingdom). Cells were imaged via Metamorph (7.10.3, MDS Analytical Technologies) by fluorescence (50–70ms exposures) with a 100×1.45 Ph3 oil objective through a 1.5 X lens, which yields 65.7 nm/pixel at bin = 1.

### Microneedle manipulation

The instruments, setup and protocol used for microneedle manipulation experiments were closely reproduced from previous work (*Suresh et al., 2020*). Computer control (Multi-Link, Sutter Instruments) was used to ensure smooth and reproducible microneedle movements. Manipulations in FCPT-treated spindles generated microneedle movements of 2.7±0.3 µm/min, consistent with previously performed wildtype spindle manipulations (2.5±0.1 µm/min) (*Suresh et al., 2020*). Cells for manipulations in FCPT-treated spindles were subjected to the same selection criteria as used in the previous study (*Suresh et al., 2020*): spindles in metaphase, flat, bipolar shape with both poles in the same focal plane.

### Data extraction, processing, and quantifications

To fit models to the data, we extracted k-fiber profiles acquired from imaging. Profile extraction was performed manually with FIJI. These profiles were rotated and aligned such that the pole and kinetochore ends are along the x-axis before model fitting. Local curvature was calculated by fitting a circle to consecutive sets of three points (spaced apart by 1 µm) along profiles and taking the inverse of the radius of the fitted circle (units=µm$^{-1}$). K-fibers were included in the data set only if their entire length stayed within the same z-plane over time, to enable accurate profile extraction. Further details on profile extraction and curvature calculation were as described in previous work (*Suresh et al., 2020*).

To distinguish between the different binding states using the equilibrium binding model (*Figure 4i*), we quantified the intensity of PRC1 and tubulin in 3 different regions: (1) across the entire spindle between the two spindle poles (not including the poles), (2) outside the spindle but inside the cell (PRC1's free population), where the cell's boundary was determined using high intensity contrast and (3) close to spindle poles (where microtubules are thought to be predominantly parallel). We averaged across multiple ROIs for (2) and (3), where the size of the ROI was kept constant (~8 pixel wide). The

measured intensity was normalized by the area of the ROIs. The chosen regions of interest for these measurements are shown in an example spindle in Appendix 5.

## Euler-Bernoulli framework for modeling k-fiber deformations

Source code for the model developed and used in this work can be found on Github (https://github.com/RPGroup-PBoC/kfiber_modeling_manipulation, *Suresh, 2022* copy archived at swh:1:rev:dee-771a47e82df5fb88c25cfc777ba377f8bb234). We adopt the Euler-Bernoulli formalism as a framework to model how k-fibers bend elastically in response to force (*Gittes et al., 1993*; *Brangwynne et al., 2006*; *Jiang and Zhang, 2008*). In this framework, curvature κ(x) at a given position x is specified through the Euler-Bernoulli equation, namely, κ(x)=- M(x)/EI. Here, M(x) is the bending moment at position x, and EI is the flexural rigidity of the k-fiber. Details on M(x) are further discussed in the Appendix 1, and the flexural rigidity EI is further discussed below.

## Flexural rigidity

Flexural rigidity (EI) is defined as the product of the elastic bending modulus (E, an intrinsic property and therefore a constant) and the areal moment of inertia (I, the second moment of inertia of the k-fiber cross section). We assume flexural rigidity of the k-fiber (EI) is constant all along its length. This is motivated by electron microscopy studies, which reveal that PtK2 cells have a large percentage of kinetochore microtubules in the k-fiber that extend all the way from the kinetochore to the pole (*McDonald et al., 1992*). This assumption allows us to report forces and moments in a ratio with EI, making our analysis independent of the precise numerical value of EI. In *Figure 2—figure supplements 1 and 3*, we report values of $M_p$ and $F_x$ as described here.

In *Figure 5h*, we report absolute forces inferred by the model. Since flexural rigidity for k-fibers has not yet been measured, we make a numerical estimate based on (1) the known number of microtubules in the k-fiber, which ranges from 15 to 25 (*McEwen et al., 1998*), (2) the known flexural rigidity of a single microtubule, $2.2 \times 10^{-23}$ $Nm^2$ (*Gittes et al., 1993*), and (3) an assumption on the strength of coupling between the microtubules in the k-fiber. The flexural rigidity of the bundle will either scale linearly with the number of microtubules (N), if the microtubules are weakly coupled and can slide with respect to each other during bending ($EI_{k-fiber}=N.EI_{MT}$), or scale quadratically with that number if the they are strongly coupled and cannot slide during bending ($EI_{k-fiber} = N^2.EI_{MT}$) (*Claessens et al., 2006*). In this work, we assume that microtubules within the k-fiber can slide ($EI_{k-fiber}=N.EI_{MT}$) and take the number of microtubules N=20. This results in a value of 400 pN.µm$^2$ for the flexural rigidity for the k-fiber, which we apply to *Figure 5h* in order to obtain absolute force estimates inferred by the minimal model.

## Modeling of native k-fiber shapes

When studying the native k-fiber shapes, we invoke the small-angle approximation ($|y'(x)|<<1$ and $κ(x) ≈ y''(x)$) which yields a second-order ordinary differential equation for the k-fiber profile y(x). This allows us to find an analytical solution for y(x) and gain insights about the role of different force contributions in dictating k-fiber shapes features (*Figure 2b*). Analytical calculations of y(x) under different scenarios and a detailed discussion of the resulting shape features can be found in Appendix 1. There, we also demonstrate the validity of the approximation by showing the agreement between its results and those obtained by a numerical solution of the exact nonlinear equation for y(x). When reporting inferred parameter values and fitting errors in *Figure 2e–f*, results of fitting the exact numerical solution of y(x) were used.

## Coarse-graining the kinetochore-proximal forces in native k-fibers

In the development of our minimal model for native k-fiber shape generation, we considered effective point forces acting at the pole and kinetochore ends of the k-fiber that were generally compressive in nature. However, it is known from prior studies (*McNeill and Berns, 1981*; *Waters et al., 1996*) that the kinetochore is under tension. An explanation of the compression present in the bulk of the k-fiber and tension in the kinetochore-proximal region was offered through the 'bridging-fiber' model (*Kajtez et al., 2016*). There, the bridging-fiber (bundle of non-kMTs in the antiparallel overlap zone) exerts a compressive force on the k-fiber at a kinetochore-proximal junction, allowing the kinetochore itself to remain under tension.

From the perspective of the bridging-fiber model, our coarse-grained treatment combined the compressive bridging-fiber force and the tensile kinetochore force into an effective compressive point force. Our simplified treatment was motivated by our goal to obtain a minimal native k-fiber model that would explain the observed shapes and serve as a foundation for incorporating anchorage forces in the microneedle manipulation studies.

To justify our coarse-graining approach, we selected a set of 32 k-fibers in their native state which had visually identifiable junctions. We then fit a more general model with a compressive bridging-fiber force at the junction and a tensile kinetochore force, inferring the forces and moments acting on the k-fiber (*Figure 2—figure supplement 1a-c*). We observed a general agreement between the inferred forces and moments between the two models (*Figure 2—figure supplement 1d*) as well as very similar RMSE values (9e-3±1e-3 µm for the junction model vs. 1e-2±1e-3 µm for our minimal model; error represents the SEM), thereby justifying the use of our minimal model.

## Modeling external force from the microneedle

The force exerted by the microneedle on the k-fiber was treated as a point force in our model. The microneedle, however, has a finite diameter, and the force it exerts is transmitted along its finite length of contact with the k-fiber. To validate the point force assumption, we simulated k-fiber profiles by considering spatially distributed microneedle forces acting along lengths ranging from 0.5 µm to 1.5 µm (*Suresh et al., 2020*). K-fiber profiles in these different settings matched each other with high accuracy when the integrated force was kept the same. In addition, when fitting a point force model to these profiles, the inferred location of the exerted point force was within ≈0.02 µm of the center of the distributed force region, and the fitting errors were very low (RMSE ≈0.03 µm). Together, these studies justify the point force assumption for the external force. A more detailed discussion of this validation study and supporting figures are included in Appendix 2.

In addition, since the manipulations are performed very slowly (average speed ≈ 0.04 µm/s, *Suresh et al., 2020*), we considered the resulting frictional/viscous force to be negligible compared to the force acting perpendicular to the k-fiber, and thus defined $\mathbf{F}_{ext}$ to be perpendicular to the tangent of the k-fiber profile.

Distributed anchorage:

To study the effect of crosslinker localization on the k-fiber's response, we mimicked the microneedle manipulation experiment synthetically for different distributions of k-fiber anchorage (*Figure 4a and c–e*). We assumed that the non-kMT network, to which the k-fiber is anchored, deforms elastically and exerts opposing forces proportional to the local deflection y(x). We model this anchorage as a series of elastic springs exerting vertical pulling forces on the k-fiber. We ignore potential contributions in the horizontal direction which could have a non-negligible effect in the case of large network deformations. Our treatment is similar to the modeling of the cellular cytoskeleton as an elastic material in earlier work (*Brangwynne et al., 2006*).

In addition, we assumed that the crosslinkers that anchor the k-fiber to the non-kMT network do not detach as a result of microneedle manipulation. If crosslinker detachment were widespread, microneedle manipulation in FCPT-treated spindles would have led to negative curvature positions occurring far away from the microneedle. We instead observed the position of negative curvature follow the microneedle, consistent with the response behavior predicted by a global anchorage scenario (*Figure 4f–h*). Based on this, we make the simplifying assumption that crosslinker detachment does not dominate the k-fiber's resistance to pivoting under manipulation.

## Modeling of k-fibers under microneedle manipulation

Since manipulated k-fiber profiles have large deflections relative to the undeformed state (|y'(x)|<<1 is not satisfied everywhere), analytical approaches for obtaining an intuitive expression for y(x) become infeasible. We therefore calculate y(x) using a numerical integration method (details in Appendix 3). Specifically, we first parameterize the k-fiber profile via an arc length parameter s and prescribe a tangential angle θ(s) to each position (*Kajtez et al., 2016*). Writing the Euler-Bernoulli equation as κ(s) = -dθ/ds = M(s)/EI and using our estimate of the local bending moment M(s) defined uniquely for each modeling scenario (*Figures 2b–4a*), we use a finite difference method to update the tangential angle at the next position s+Δs. Steps in the x- and y-directions are then performed using the updated tangential angle.

## Assumption of a uniform flexural rigidity

To test whether the naturally occurring nonuniformity of k-fibers (*McDonald et al., 1992*; *O'Toole et al., 2020*) may be sufficient to explain the presence of negative curvature minima within 3 µm from the kinetochore (*Figure 3h*), we performed a set of simulation studies involving k-fibers with nonuniform thickness. Specifically, we considered k-fibers that had γ times larger flexural rigidity (EI) in their kinetochore-proximal region of length s*. Applying a vertical external force and a point negative bending moment $M_k$ at the kinetochore, we studied the k-fiber response. We first tested the impact of having non-uniform k-fiber thickness on the negative curvature response for an extreme scenario with *γ*=3. Together with a point negative moment at the kinetochore, this was sufficient to elicit a shift in the negative curvature minimum away from the kinetochore (*Figure 3—figure supplement 2a,b*).

To find out whether such behavior could be observed for realistic choices of γ, we performed a parameter study where we systematically tuned the γ parameter along with the negative bending moment at the kinetochore. For each pair of parameters, we calculated the ratio of negative curvature values at the thickness transition point (κ*) and at the kinetochore ($κ_K$). Values of this ratio (κ*/$κ_K$) that were greater than 1 indicated the presence of the curvature minimum at the transition point (away from the kinetochore), while values less than 1 meant the curvature minimum was at the kinetochore (*Figure 3—figure supplement 2c*).

Our study revealed that a nonuniform EI model with a realistic γ value (1.3) can, in fact, result in negative curvature minima within 0.5 µm from the kinetochore (*Figure 3—figure supplement 2c*, left). However, the k-fiber would need to be at least twice as thick near the kinetochore to have the negative curvature minimum be 1.5 µm away from the kinetochore. This requirement is increased to 3.5 times if the curvature minimum is to be 2.5 µm away from the kinetochore (*Figure 3—figure supplement 2c*, right). The minimum required γ parameter values for other choices of s* are shown in *Figure 3—figure supplement 2d*.

Overall, this study suggests that while 20–30% higher thickness of the k-fiber in the kinetochore-proximal region (*McDonald et al., 1992*; *O'Toole et al., 2020*) may contribute to k-fiber's resistance to bending, it cannot be the dominating factor eliciting the negative curvature response.

## Model fitting and error estimation

In our model fitting procedure, we minimize the sum of squared errors. For a given data point ($x_i$, $y_i$) on the tracked k-fiber, we define the error as the minimal distance between that point and the k-fiber profile predicted by the model. If the point lies exactly on the predicted profile, the corresponding error will be zero.

We obtain the optimal set of model parameters through a combination of deterministic least-squares minimization and stochastic search algorithms initialized at multiple different locations in parameter space. This is done to prevent the method from converging to a local optimum. During parameter search, we impose constraints on the parameter values to prevent the realization of unphysical configurations. These constraints are that the k-fiber profile cannot form loops, the inferred external force necessarily points outward, and forces of k-fiber end-points are lower than the critical buckling force. In addition, due to the uncertainties associated with precisely determining the positions of the microneedle contact, we let our search method consider positions within 0.5 µm of prescribed values. Details on estimating the fitting error and finding optimal model parameters are included in Appendix 4.

## Modeling the binding states of PRC1

We calculate the free, singly bound and doubly bound populations of PRC1 using equilibrium thermodynamic modeling combined with the measured immunofluorescence of PRC1 and tubulin within the spindle. The free PRC1 population was estimated using measured intensities in intracellular regions with very low tubulin presence. Then, the free ($c_f$) and singly bound ($c_1(\mathbf{r})$) populations were related via $c_1(\mathbf{r}) = \rho_{MT}(\mathbf{r})c_f/K_d$, where $\rho_{MT}(\mathbf{r})$ is the local tubulin concentration. The dissociation constant $K_d$ was inferred from the PRC1 and tubulin concentrations (measured in arbitrary units) in the pole-proximal regions of the spindle, where microtubules are known to be predominantly parallel (*Euteneuer and McIntosh, 1981*). The doubly bound PRC1 population ($c_2(\mathbf{r})$) that contributes to k-fiber crosslinking was then obtained by subtracting the free and singly bound contribution from the measured total population. In *Figure 4i*, the concentration of actively engaged crosslinkers per tubulin, i.e., $c_2(r)$/

$\rho_{MT}(\mathbf{r})$, was reported along the pole-pole axis. More details on the methodology of separating the binding states of PRC1 are provided in Appendix 5.

## Quality of fit assessments and statistical analyses

When comparing the quality of fits between different modeling scenarios, we report the average root-mean-squared error (RMSE) values, along with the standard error of the mean (SEM) calculated for each scenario (*Figures 2e, 3f and 5c*).

We report other metrics for assessing the quality of fits in which we compare different signature shape features between the tracked profile and the model predicted profile. For the native k-fiber model scenarios, this includes the location of the peak deflection (*Figure 2d and f*). For manipulated k-fiber model scenarios, these include the location of curvature maximum (*Figures 3g and 5d*), the location of curvature minimum (*Figures 3h and 5e*), and the length over which k-fiber orientation is strictly preserved within 1° (*Figures 3i and 5f*).

We used the non-parametric two-sided Mann-Whitney U test when comparing two independent datasets and display the p-values on the figures (*Figures 2e, 3f and 5c*). In the text, each time we state a significant change or difference, the p-value for those comparisons were less than 0.05. To evaluate the correlations between the data and model (such as the comparison of signature shape features), we used the Pearson correlation function to test for linearity (*Figures 3g–i, 5d–f and h*). We report the coefficient of determination, $R^2$, which assesses how well the model captures the variance in the features of interest observed in the data. To test for monotonic relationships between two variables (*Figure 5i*), we used the Spearman correlation function. In the legends we state what test was conducted. Quoted m's refer to the number of individual cells and n's refer to the number of individual k-fibers.

## Acknowledgements

We thank Alexey Khodjakov for PtK2 GFP-α-tubulin cells and Timothy Mitchison for FCPT. We are grateful to Nenad Pavin for helpful discussions, and Arthur Molines, Soichi Hirokawa, Miquel Rosas Salvans, Lila Neahring, Caleb Rux, Gabe Salmon, and other members of the Phillips and Dumont Labs for critical feedback on our work. This work was supported by NIH 1R01GM134132, NIH R35GM136420, NSF CAREER 1554139, NSF 1548297 Center for Cellular Construction (SD), NIH 2R35GM118043-06, the John Templeton Foundation 51250 and 60973 (RP), the Chan Zuckerberg Biohub (SD and RP), NSF Graduate Research Fellowship and UCSF Kozloff Fellowship (PS).

## Additional information

### Funding

| Funder | Grant reference number | Author |
| --- | --- | --- |
| National Institutes of Health | 1R01GM134132 | Sophie Dumont |
| National Institutes of Health | R35GM136420 | Sophie Dumont |
| National Science Foundation | 1554139 | Sophie Dumont |
| National Science Foundation | 1548297 | Sophie Dumont |
| John Templeton Foundation | 51250 | Rob Phillips |
| John Templeton Foundation | 60973 | Rob Phillips |
| National Institutes of Health | 2R35GM118043-06 | Rob Phillips |

| Funder | Grant reference number | Author |
|---|---|---|
| Chan Zuckerberg Initiative | | Rob Phillips |
| National Science Foundation | GRFP | Pooja Suresh |

The funders had no role in study design, data collection and interpretation, or the decision to submit the work for publication.

## Author contributions

Pooja Suresh, Conceptualization, Data curation, Software, Formal analysis, Funding acquisition, Validation, Investigation, Visualization, Methodology, Writing - original draft, Project administration, Writing – review and editing; Vahe Galstyan, Conceptualization, Data curation, Software, Formal analysis, Validation, Investigation, Visualization, Methodology, Writing - original draft, Project administration, Writing – review and editing; Rob Phillips, Conceptualization, Resources, Supervision, Funding acquisition, Project administration, Writing – review and editing; Sophie Dumont, Conceptualization, Resources, Funding acquisition, Project administration, Writing – review and editing

### Author ORCIDs
Pooja Suresh ⓘ http://orcid.org/0000-0002-7793-4827
Vahe Galstyan ⓘ http://orcid.org/0000-0001-7073-9175
Rob Phillips ⓘ http://orcid.org/0000-0003-3082-2809
Sophie Dumont ⓘ http://orcid.org/0000-0002-8283-1523

### Decision letter and Author response
Decision letter https://doi.org/10.7554/eLife.79558.sa1
Author response https://doi.org/10.7554/eLife.79558.sa2

## Additional files

### Supplementary files
• MDAR checklist

### Data availability
All microscopy images (data) used for our study is available on github (https://github.com/RPGroup-PBoC/kfiber_modeling_manipulation, copy archived at swh:1:rev:dee771a47e82df5fb88c25cfc777ba377f8bb234).

The following datasets were generated:

| Author(s) | Year | Dataset title | Dataset URL | Database and Identifier |
|---|---|---|---|---|
| Suresh P, Galstyan V, Phillips R, Dumont S | 2022 | Microneedle manipulation in FCPT-treated Ptk2 cells (metaphase spindles) | https://github.com/RPGroup-PBoC/kfiber_modeling_manipulation/tree/main/dat/FCPT_manipulated_k_fiber_shapes | github, FCPT_manipulated_k-fiber_shapes |
| Suresh P, Galstyan V, Phillips R, Dumont S | 2022 | Unmanipulated metaphase spindles in Ptk2 cells | https://github.com/RPGroup-PBoC/kfiber_modeling_manipulation/tree/main/dat/WT_unmanipulated_k-fiber_shapes | github, WT_unmanipulated_k-fiber_shapes |

The following previously published datasets were used:

| Author(s) | Year | Dataset title | Dataset URL | Database and Identifier |
|---|---|---|---|---|
| Suresh P, Galstyan V, Phillips R, Dumont S | 2020 | Immunofluorescence images for PRC1 localization in metaphase spindles of Ptk2 cells | https://github.com/RPGroup-PBoC/kfiber_modeling_manipulation/tree/main/dat/PRC1_localization | github, PRC1_localization |
| Suresh P, Galstyan V, Phillips R, Dumont S | 2020 | Microneedle manipulation in control Ptk2 cells (metaphase spindles) | https://github.com/RPGroup-PBoC/kfiber_modeling_manipulation/tree/main/dat/WT_manipulated_k-fiber_shapes | Github, WT_manipulated_k-fiber_shapes |

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

## Appendix 1

## Euler-Bernoulli formalism and generation of native k-fiber shapes

In this section, we use the Euler-Bernoulli formalism to calculate the shapes of native k-fiber profiles (see *Figure 2c* of the main text for example profiles). In our analysis, we consider k-fibers at mechanical equilibrium and assume that their shapes are generated by forces and moments acting at their end-points. All calculations are performed in the reference frame where the pole and the kinetochore lie along the $x$-axis. The calculations and results of this section are related to *Figure 2* of the main text.

### 1.Euler-Bernoulli equation and the small angle approximation

The Euler-Bernoulli beam theory relates the local curvature $\kappa(x)$ to the bending moment $M(x)$ via

$$\kappa(x) = -\frac{M(x)}{EI}, \tag{1}$$

where $EI$ is the flexural rigidity of the beam, with $E$ being the Young's modulus, and $I$ being the areal moment of inertia. The general expression for the curvature written in terms of Cartesian coordinates is given by

$$\kappa(x) = \frac{y''(x)}{\left(1 + y'(x)^2\right)^{3/2}}. \tag{2}$$

When substituted into *Equation 1*, this expression for $\kappa(x)$ results in a nonlinear equation for the k-fiber profile $y(x)$, making it challenging to obtain an analytical solution and extract intuition from it. We therefore begin our calculations by making the so-called 'small angle approximation' in order to write $\kappa(x) \approx y''(x)$. This approximation applies to the native k-fibers shapes, which, upon aligning them along the $x$-axis, appear flat and have a small tangential angle at every position of the profile (i.e., $|y'(x)| \ll 1$). This leads to a simpler and analytically tractable form for the Euler-Bernoulli equation, namely

$$y''(x) = -\frac{M(x)}{EI}. \tag{3}$$

Solving this simpler equation will let us gain insights into how the different model components that define $M(x)$ uniquely contribute to k-fiber shape. Later in Appendix section 1.3, we will demonstrate the validity of applying the 'small angle approximation' for native k-fiber shapes.

### 2.Analytical solutions for native k-fiber profiles

In our minimal model, the shape of native k-fibers is generated due to forces and moments acting at the pole and kinetochore ends of the k-fiber (*Figure 2b* of the main text). The $x$- and $y$-components of the force at the pole are denoted by $F_x$ and $F_y$, respectively. The bending moment at the pole is denoted by $M_p$, with the counterclockwise direction chosen to be positive. The bending moment at the kinetochore ($M_k$) is generally different from $M_p$.

*Equation 3* is a second-order ordinary differential equation for the k-fiber shape $y(x)$. To obtain $y(x)$, we need to specify two boundary conditions. These condition are

$$y(0) = 0, \tag{4}$$

$$y(L) = 0, \tag{5}$$

where $L$ is the distance between the pole and kinetochore ends of the k-fiber. These conditions require that the k-fiber ends are positioned on the $x$-axis.

The local bending moment $M(x)$ is obtained by writing the moment balance condition for the $[0, x]$ segment of the k-fiber (see *Figure 2b* of the main text). Specifically, $M(x)$ needs to balance the torque generated by the force $\vec{F} = (F_x, F_y)$ at the pole and the bending moment $M_p$, i.e.,

$$M(x) = M_p + \underbrace{F_x\, y(x) - F_y\, x}_{\left(\vec{r} \times \vec{F}\right)_z}. \tag{6}$$

Using the above expression, we can relate the bending moment at the kinetochore ($M_k \equiv M(x = L)$) to the moment at the pole ($M_p$). Noting that $y(L) = 0$, we obtain

$$M_k = M_p - F_y L. \tag{7}$$

This indicates that a non-zero end-point force perpendicular to the pole-kinetochore axis will necessarily result in different bending moments (hence, curvatures) at k-fiber ends.

Knowing how the bending moment varies in space (*Equation 6*), we now substitute it into the Euler-Bernoulli equation in its 'small angle approximation' form (*Equation 3*) and obtain a linear second order ODE for the profile function $y(x)$, namely

$$y''(x) + \left(\frac{F_x}{EI}\right) y(x) = -\frac{M_p}{EI} + \left(\frac{F_y}{EI}\right) x. \tag{8}$$

We note that the forces and the moment at the pole always appear in a ratio with the flexural rigidity $EI$. We therefore introduce rescaled effective parameters $\tilde{F}_x = F_x/EI$, $\tilde{F}_y = F_y/EI$, and $\tilde{M}_p = M_p/EI$, and rewrite the second order ODE for $y(x)$ as

$$y''(x) + \tilde{F}_x y(x) = -\tilde{M}_p + \tilde{F}_y x. \tag{9}$$

The functional form of the solution for $y(x)$ depends on the signs and values of the different parameters. Therefore, in the following, we consider separate scenarios and discuss the insights that each analytical solution provides.

$F_x = 0$. We begin with the special case where the point force along the pole-kinetochore axis is zero. This simplifies the differential equation into

$$y''(x) = -\tilde{M}_p + \tilde{F}_y x. \tag{10}$$

Integrating twice over $x$, we obtain

$$y(x) = C_1 + C_2 x - \frac{\tilde{M}_p x^2}{2} + \frac{\tilde{F}_y x^3}{6}, \tag{11}$$

where $C_1$ and $C_2$ are integration constants. Imposing the boundary conditions (*Equation 4* and *Equation 5*), we find these constants to be $C_1 = 0$ and $C_2 = \tilde{M}_p L/2 - \tilde{F}_y L^2/6$. Substituting $C_1$ and $C_2$, and writing the perpendicular force as $\tilde{F}_y = (M_p - M_k)/L$ (from *Equation 7*), we obtain the final expression for $y(x)$:

$$y(x) = \tilde{M}_p \underbrace{\frac{x(L - x)}{2}}_{\text{symmetric}} - (\tilde{M}_p - \tilde{M}_k) \underbrace{\frac{x(L^2 - x^2)}{6L}}_{\text{asymmetric}}. \tag{12}$$

The first term contributing to the profile is symmetric about the middle position $x = L/2$ and does not change under the transformation $x \to L - x$. The second term, however, is asymmetric and leads to a shift of the profile peak toward the end which has the higher bending moment.

In the limit where there is bending moment at the pole but not at the kinetochore ($\tilde{M}_p > 0$, $\tilde{M}_k = 0$), we can find the peak position of the asymmetric profile by solving for $x$ in the equation $y'(x) = 0$. We obtain $x_{\text{peak}} = \left(1 - \sqrt{3}/3\right) L \approx 0.42\,L$, which means that the peak of the profile is shifted toward the pole side by $\approx 8\%$ of the end-to-end distance $L$. Similarly, when bending is present only at the kinetochore ($\tilde{M}_p = 0$, $\tilde{M}_k > 0$), the profile peaks at $x \approx 0.58\,L$, which is shifted now toward the kinetochore side by the same amount (see *Figure 2c* of the main text for demonstrations of these two asymmetric cases).

$F_x > 0$ and $M_p = M_k = 0$. Next, we consider another special case where the k-fiber profile is formed by a purely axial force $F_x$ in the absence of bending moments at either end-point ($\tilde{M}_p = \tilde{M}_k = 0$ and hence, $\tilde{F}_y = 0$). The ODE for $y(x)$ in this case simplifies into

$$y''(x) + \tilde{F}_x\,y(x) = 0. \tag{13}$$

The general solution is a linear combination of $\sin(kx)$ and $\cos(kx)$ functions, with the wave number defined as $k = \sqrt{\tilde{F}_x}$. The boundary condition $y(0) = 0$ eliminates the cosine solution. Imposing the

second boundary condition, we obtain $\sin(kL) = 0 \Rightarrow k = \pi/L$ (first buckling mode). This suggests that the axial force needs to exactly equal the critical buckling force given by $F_c = \pi^2 EI/L^2$. The sinusoidal buckling profile, as shown in **Figure 2c**, is symmetric with respect to $x = L/2$.

$F_x > 0$, $M_p > 0$, and $M_k = 0$. We end our analytical treatment of native k-fiber shapes by considering the more general case where a moment at the pole and axial forces are both present, but there is no moment generation at the kinetochore ($\tilde{M}_k = 0$). This corresponds to the minimal model sufficient to capture the diverse shapes of k-fibers in their native state (see **Figure 2f and g** of the main text).

Substituting $\tilde{F}_y = \tilde{M}_p/L$ into **Equation 9**, the ODE for the k-fiber profile $y(x)$ for this case becomes

$$y''(x) + \tilde{F}_x y(x) = -\frac{\tilde{M}_p}{L}(L - x). \tag{14}$$

The general solution to the ODE can be written as

$$y(x) = D_1 \sin\left(k(L - x)\right) + D_2 \cos\left(k(L - x)\right) - \frac{\tilde{M}_p}{\tilde{F}_x L}(L - x), \tag{15}$$

where $D_1$ and $D_2$ are integration constants. The argument of sine and cosine functions is written as $(L - x)$ for convenience. Now, the boundary condition at the kinetochore is $y(x = L) = 0$ which indicates that $D_2 = 0$. From a similar boundary condition at the pole ($y(0) = 0$), we obtain $D_1 = (\tilde{M}_p/\tilde{F}_x)/\sin(kL)$. After substitution, the final expression for the profile becomes

$$y(x) = \frac{\tilde{M}_p}{\tilde{F}_x}\left(\frac{\sin(k(L - x))}{\sin(kL)} - \frac{L - x}{L}\right), \tag{16}$$

where, as a reminder, $k = \sqrt{\tilde{F}_x}$. One can show that in the limit where the axial force goes to zero ($k \to 0$), the polynomial solution in **Equation 12** is recovered. Conversely, when the axial force is close to the critical buckling force (achieved when $kL \approx \pi$ or $\tilde{F}_x \approx \pi^2/L^2$), the sine term becomes dominant in the solution and the symmetric sinusoidal profile is recovered (solution of **Equation 13**).

To probe the behavior in the intermediate regimes, we tuned the axial force in the $(0, F_c)$ range and numerically found the corresponding moment at the pole ($M_p$) that would yield an identical peak deflection, which we set equal to $y_{max} = 0.1L$ (our conclusions hold true for any other $y_{max}$ value that does not violate the small-angle approximation). As shown in **Appendix 1—figure 1a**, the larger the axial force becomes, the smaller the corresponding moment at the pole needs to be in order to yield the same amount of k-fiber deformation (measured by $y_{max}$). Furthermore, as anticipated, increasing the axial force (with a corresponding decrease in the moment at the pole) shifts the position of the peak closer to the center (**Appendix 1—figure 1b**), and when the force reaches the critical buckling force, the $x$-position of the peak becomes equal to $L/2$.

**(a)**                                                    **(b)**

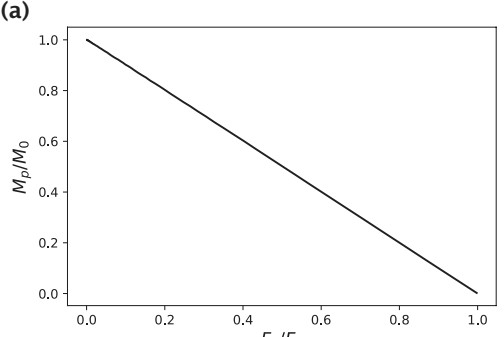                                       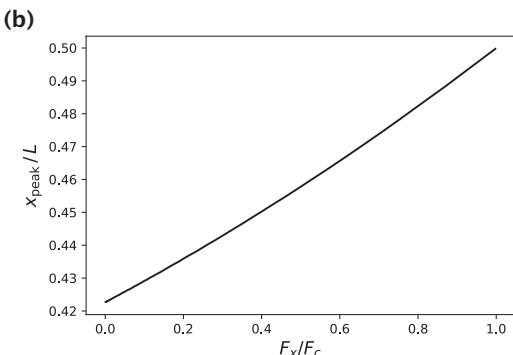

**Appendix 1—figure 1.** Generation of native k-fiber shapes with a fixed peak deflection $y_{max}$ for different choices of the axial force $F_x$. (**a**) Moment at the pole $M_p$ that yields the specified peak deflection $y_{max}$ as a function of $F_x$. $M_0$ is the pole moment in the absence of an axial force. (**b**) $x$-position of the profile peak as a function of $F_x$.

Overall, this study shows that the simultaneous tuning of $M_p$ and $F_x$ (equivalently, $F_y$ and $F_x$) according to the rule revealed in **Appendix 1—figure 1a** shifts the $x$-position of the k-fiber peak without changing the magnitude of the peak k-fiber deflection ($y_{max}$).

## 3.Justification of the small angle approximation

Here, we demonstrate the validity of the small angle approximation for modeling native k-fiber shapes by comparing the results of inference under this approximation with the results of the more exact numerical approach detailed in Appendix section 3.

In our minimal model of native k-fiber shape generation, the two independent parameters are the axial force $F_x$ and the moment at the pole $M_p$. For each parameter, we take the ratio of its inferred value under the approximate and exact methods, and plot the value of this ratio for all native k-fibers considered in our study (*Appendix 1—figure 2a*). When plotting, we give transparency to each data point based on how much they contribute to k-fiber shape. If the data point is transparent, then the corresponding parameter ($F_x$ or $M_p$) contributes little to k-fiber shape. We set the weights of shape contribution for axial force and moment at the pole as $w_{F_x} = |F_x|y_{\max}/(|F_x|y_{\max} + M_p)$ and $w_{M_p} = M_p/(|F_x|y_{\max} + M_p)$, respectively, where $|F_x|y_{\max}$ is the largest mechanical moment exerted by the axial force about the origin $(0,0)$. As can be seen from *Appendix 1—figure 2a*, parameters inferred by the two methods are almost always very close to each other when the corresponding parameter has a significant shape contribution, and may differ significantly when the corresponding parameter does not contribute significantly to shape (transparent points). Furthermore, the fitting errors predicted by the two methods are very similar to each other (*Appendix 1—figure 2b*). Together, these studies demonstrate the validity of invoking the small angle approximation for studying native k-fiber shapes.

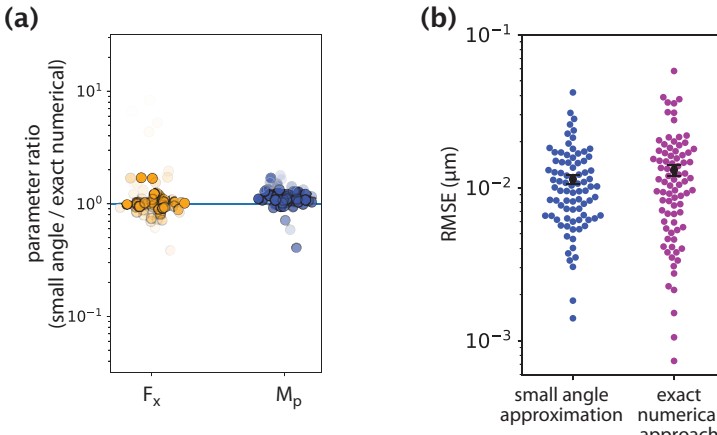

**Appendix 1—figure 2.** Comparison of inference results from the small angle approximation and exact numerical approach. (**a**) Ratios of parameters inferred by the two methods. (**b**) Fitting error comparison between the two methods.

## 4. Dependence of the largest normalized deflection on native k-fiber length

Here, we use the analytical solutions of native k-fiber profiles obtained under the small angle approximation (Appendix section 1.2) to study the dependence of the largest normalized $y$-deflection on the length of the k-fiber when the force parameters are kept fixed.

We begin with the special case where the k-fiber profile is generated by a point moment $\tilde{M}_p$ acting at the pole. In this case, the profile $y(x)$ is given via

$$y(x) = \tilde{M}_p \frac{x(L-x)(2L-x)}{6L}, \tag{17}$$

which follows from *Equation 12* by setting $\tilde{M}_k = 0$. Earlier in Appendix section 1.2, we found that the profile peak was achieved at $x_{\text{peak}} = \left(1 - \sqrt{3}/3\right) L$. Substituting this $x$-value into *Equation 17* and simplifying the resulting expression, we obtain the largest $y$-deflection, namely

$$y_{\max} = \frac{\sqrt{3}}{27} \tilde{M}_p L^2. \tag{18}$$

We are interested in the largest $y$-deflection normalized by the end-to-end distance $L$. Dividing both sides of **Equation 18** by $L$, we find

$$\frac{y_{\max}}{L} = \frac{\sqrt{3}}{27}\tilde{M}_p L. \tag{19}$$

As we can see, the normalized deflection of the bent k-fiber profile scales linearly with the end-to-end distance $L$, with the slope of the dependence given by $(\sqrt{3}/27)\tilde{M}_p$.

When the k-fiber profile is generated by the combined action of a moment at the pole ($\tilde{M}_p$) and an axial force ($\tilde{F}_x$), the normalized k-fiber profile follows from our earlier derived result for $y(x)$ in **Equation 16**, namely

$$\frac{y(x)}{L} = \frac{\tilde{M}_p}{\tilde{F}_x L}\left(\frac{\sin(k(L-x))}{\sin(kL)} - \frac{L-x}{L}\right), \tag{20}$$

where $k = \sqrt{\tilde{F}_x}$. An exact analytical solution for $y_{\max}/L$ is available but is not informative due to its complexity. We therefore use a numerical approach to study the more general case.

We first calculated deflection profiles for k-fibers of different lengths by fixing the moment at the pole ($\tilde{M}_p$) and the axial force ($\tilde{F}_x$) at their average values inferred for native k-fibers (**Appendix 1—figure 3a**). As expected, longer k-fibers are deformed more when exposed to the same end-forces. The increasing trend also holds true when we normalize the axes by the end-to-end distance $L$ (**Appendix 1—figure 3b**). This corresponds to the trend we observed in the normalized deflections measured for inner vs. outer native k-fibers (**Figure 2—figure supplement 2b**).

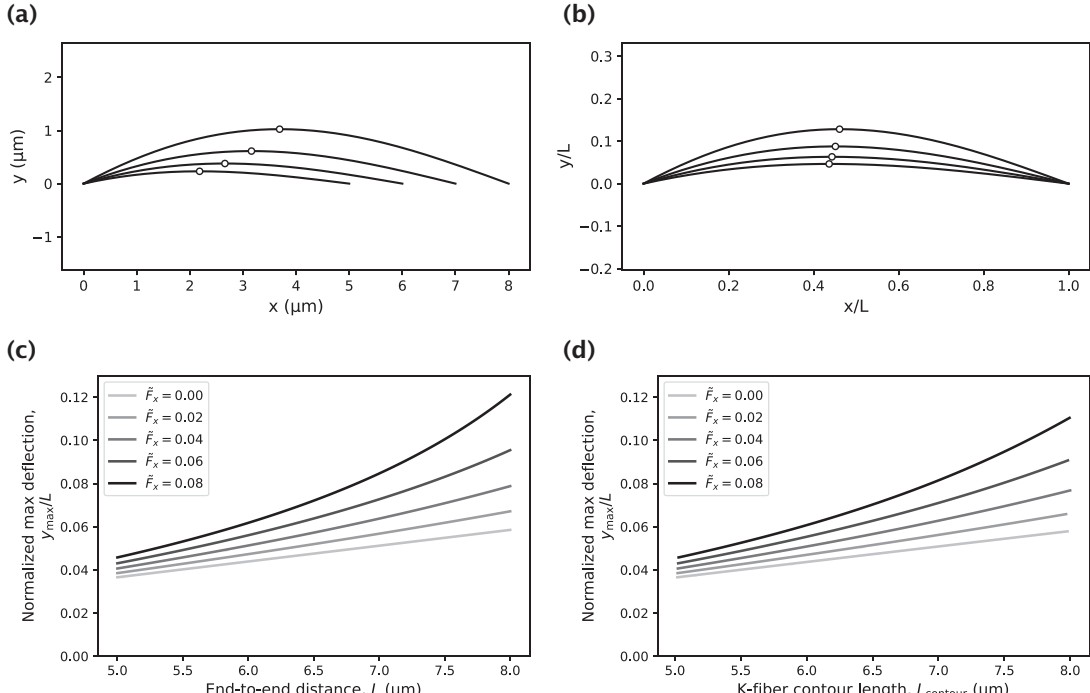

**Appendix 1—figure 3.** Impact of the k-fiber length on the amount of k-fiber deflection. (**a**) Set of k-fiber profiles of varying contour length generated by a moment at the pole ($M_p = 0.114\,EI\,\mu\mathrm{m}^{-1}$) and an axial force ($F_x = 0.084\,EI\,\mu\mathrm{m}^{-2}$). Circles indicate the locations of the highest deflection. (**b**) K-fiber profiles in panel (**a**), normalized by the end-to-end distance $L$. (**c**) Dependence of the normalized maximum deflection on the end-to-end distance $L$ for varying choices of the axial force $F_x$ (reported in units of $EI\,\mu\mathrm{m}^{-2}$) and a fixed value of the moment at the pole, $M_p = 0.114\,EI\,\mu\mathrm{m}^{-1}$. (**d**) Study in panel (**c**) with the $x$-axis corresponding to the k-fiber contour length $L_{\mathrm{contour}}$ instead of the end-to-end distance $L$.

To better understand the effect that the inclusion of axial forces has on the maximum deflection, we plotted the dependence of $y_{\max}/L$ on the end-to-end distance $L$ for different choices of the axial force $\tilde{F}_x$ (**Appendix 1—figure 3c**). In the limiting case where $\tilde{F}_x = 0$, the linear scaling derived in

*Equation 19* is recovered. With increasing values of $\tilde{F}_x$, the dependence on $L$ becomes stronger and the curves shift upward. This indicates that in the presence of axial forces, a wide range of deflection amounts can be achieved by tuning the k-fiber end-to-end distance in its physiological range.

We end this section by noting that the effects captured in *Appendix 1—figure 3c* are closely observed if instead of tuning the end-to-end distance, we tuned the k-fiber contour length instead (*Appendix 1—figure 3d*). This is because for small deflections, those two length measures are almost identical.

# Appendix 2

## Justification of treating the microneedle force as a point force

In this section, we present the results of the studies that justify our treatment of the microneedle force as a point force in models of k-fiber shape generation (related to *Figures 3–5* in the main text). This is in contrast to using a distributed force acting along a finite region of the k-fiber, the size of which is set by the diameter of the microneedle (≈0.5–1.5 μm).

To demonstrate that k-fiber response is not significantly dependent on the distributed force assumption, we first generated k-fiber shapes by applying distributed forces over regions of varying size (0.5–1.5 μm) and overlaid the resulting profiles. As shown in *Appendix 2—figure 1*, there is a very close match between the k-fiber profiles generated by the same integrated force distributed along different length scales.

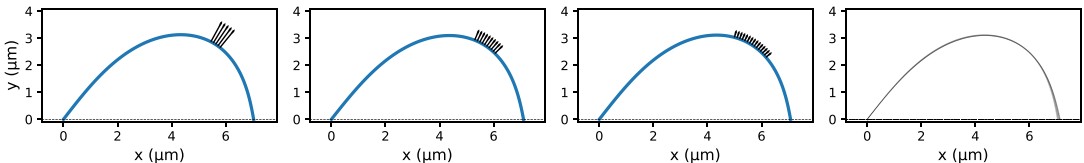

**Appendix 2—figure 1.** K-fiber profile generation with a distributed microneedle force. (**a**) Example profiles where the same integrated microneedle force was applied over three different regions of the k-fiber. The black arrows indicate the applied external forces with lower magnitudes for larger regions of distributed force application. Parameters used in profile generation: $\tilde{F}_{\text{ext}} = 0.2\,\mu\text{m}^{-2}$ (integrated force), $L_{\text{contour}} = 10$ μm. (**b**) Profiles from 10 different distributed force settings overlaid on top of each other. Only minor differences can be observed near the rightmost end.

To further validate our point-force treatment, we inferred a point-force model for the synthetically generated profiles with distributed forces (see Appendix sections 3 and 4 for details on the inference procedure). In all cases, the effective point force was inferred to be within ≈ 0.02 μm of the center of the distributed force application region (*Appendix 2—figure 2*), with the error between the generated data and inferred model being very low (RMSE ≈ 0.03 μm). Together, these results justify our treatment of the microneedle force as a point force in our models of k-fiber manipulation.

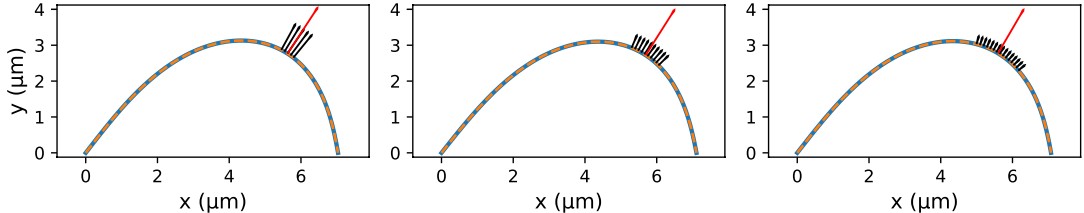

**Appendix 2—figure 2.** Fits of the point microneedle force model (dashed lines) to synthetic profiles (solid blue lines) which were generated with distributed forces applied over different regions. The red arrow indicates the inferred point microneedle force $F_{\text{ext}}$.

## Appendix 3

### Numerical approach for solving the Euler-Bernoulli equation

In this section, we present in detail the approach we took to numerically solve the Euler-Bernoulli equation in the general scenario of arbitrary deformation magnitudes where the small angle approximation applied in the previous section may no longer be valid. The approach detailed in this section as well as the following one is relevant to *Figures 2–5* in the main text.

Since in general k-fiber profiles may have more than one $y$-intercept at a given $x$ position, we consider a parameterization of the k-fiber shape via an arc length parameter $s$ and solve for $(x(s), y(s))$ instead. Assigning a tangential angle $\theta(s)$ to each point on the k-fiber, we write the Euler-Bernoulli equation as

$$\kappa(s) = -\frac{\mathrm{d}\theta}{\mathrm{d}s} = \frac{M(s)}{EI}. \tag{21}$$

As we can see, knowing the bending moment $M(s)$ at a given arc length position $s$ gives us the rate of change in the tangential angle $\theta(s)$.

To numerically solve for the k-fiber profile, we need initial conditions and update rules. For convenience, we always initialize the first k-fiber point at the origin and therefore set $x(0) = y(0) = 0$. Euler-Bernoulli equation provides us with the derivative of the tangential angle $\theta(s)$. The initial angle $\theta(0)$ therefore needs to be initialized also. When fitting the model to experimental data, we make an educated guess for $\theta(0)$ based on the initial tangential angle of the data profile. Over the course of model optimization, this initial angle is treated as a parameter and is optimized over for better model fitting.

The k-fiber shape profile is solved iteratively using a finite difference method. Specifically, the bending moment $M(s)$ is first used to estimate the tangential angle at position $s + \Delta s/2$ via

$$\theta(s + \Delta s/2) = \theta(s) - \frac{M(s)}{EI}\frac{\Delta s}{2}. \tag{22}$$

Then, the new coordinate on the k-fiber profile is calculated using this angle via

$$x(s + \Delta s) = x(s) + \Delta s \, \cos(\theta(s + \Delta s/2)), \tag{23}$$

$$y(s + \Delta s) = y(s) + \Delta s \, \sin(\theta(s + \Delta s/2)). \tag{24}$$

Finally, the tangential angle is updated for the next step using the bending moment at $s + \Delta s/2$, namely

$$\theta(s + \Delta s) = \theta(s) - \frac{M(s + \Delta s/2)}{EI}\Delta s. \tag{25}$$

Our approach of dividing each step into two half-steps reduces the error in shape calculation, making it quadratic in the step size, i.e., $O(\Delta s^2)$.

For each separate model considered in our work (*Figure 2a*, *Figure 3b and d*, *Figure 4b*, and *Figure 5a*), the corresponding expression for the bending moment is used when evaluating $M(s) = M(\vec{r}(s))$. Moment contribution from the microneedle force $\vec{F}_{\mathrm{ext}}$ is considered only when the current position $x(s)$ exceeds the position of the external force $x_{\mathrm{ext}}$. A similar treatment is used also for the point crosslinking force $\vec{F}_c$ applied at position $x_c$. Lastly, in the case of distributed crosslinking shown in *Figure 4b*, we account for the integrated moment contribution if the current position $x(s)$ falls in the crosslinking region $(L - \sigma, L)$. Mechanical moments of distributed crosslinking forces up to position $x(s)$ are calculated by treating them as a series of linear Hookean springs exerting a restoring force $-ky(s')\Delta x'$ on k-fiber segments $(s', s' + \Delta s)$ with an $x$-projection size $\Delta x'$. Here, $k$ is the effective 'spring constant' chosen in our studies to be sufficiently large to result in a negative curvature response near the kinetochore.

## Appendix 4

## Model fitting procedure

We obtain the best model fits to the extracted k-fiber profiles by minimizing the sum of squared errors. Error is defined for each data point of the tracked k-fiber as the minimal distance between that point and the model k-fiber profile, with the model profile represented as a piecewise linear curve (*Appendix 4—figure 1*). This error metric can be successfully applied to profiles with sophisticated shapes, as opposed to the more traditional metrics based on errors in $y(x)$ prediction which become ill-defined for curved profiles with more than one $y$-value at a given $x$-position.

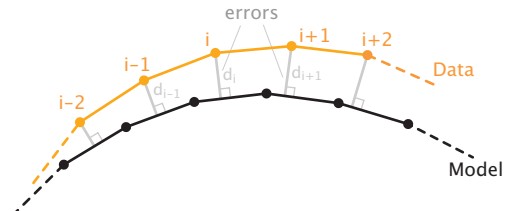

**Appendix 4—figure 1.** Schematic of the error definition. For the $i^{\text{th}}$ data point, the error $d_i$ is the smallest distance to the model profile which is represented as a piecewise linear curve. Sum of squared errors, namely, $\sum_i d_i^2$, is minimized in the model fitting procedure.

During parameter search, we impose constraints on the parameter values and possible k-fiber shape profiles based on our understanding of the experimental setup. This helps us avoid physically unrealistic scenarios. Below we list the main constraints that we imposed:

- The inferred microneedle force has to point outward, with its $y$-component having the same sign as k-fiber deflection, i.e., it has to be positive.
- The inferred microneedle force is perpendicular to the local tangent of the k-fiber shape profile. This is based on our assumption of very low frictional forces in the tangent direction discussed in the Materials and Methods.
- The inferred position of microneedle force application is within 0.5 μm of the position of profile peak. We give this finite range in our search for the effective point of force application because the precise point of contact between the k-fiber and the microneedle is hard to identify from fluorescence microscopy images, and because the contact likely takes place over a small but finite contour length.
- If the microneedle force has a positive $x$-component (points to the right), then the point force at the left k-fiber endpoint ($F_x$) has to have a negative $x$-component (point to the left) in order to balance the external force. This condition is imposed to avoid considering spontaneous inward-pointing buckling forces during parameter search. An analogous constraint is applied if the microneedle force has a negative $x$-component (points to the left).
- Parameter sets that predict k-fiber shape profiles with loops (i.e., the model curve passes through the same point twice) are not considered during search.

Because of this set of hard constraints, standard gradient descent-based methods for minimizing the sum of squared errors are not effective for finding the optimal parameters. We therefore use a time consuming but more reliable stochastic search method. There, we initialize multiple "walkers" at different positions in the parameter space, and run a stochastic search with up to 150,000 steps where each step is more likely to be taken in the direction that decreases the overall error. The best model fit is then generated by the set of sampled parameters that yields the lowest sum of squared errors.

## Appendix 5

### Modeling of PRC1 binding states

In this section, we provide the details of our approach to distinguish the PRC1 populations by their binding state using equilibrium thermodynamic modeling and immunofluorescence imaging data (*Suresh et al., 2020*). The results presented here are relevant to *Figure 4i* of the main text.

We distinguish three binding states for PRC1 - freely diffusing, singly bound, and doubly bound. The freely diffusing population represents the unbound PRC1 molecules that occupy the entire volume of the cell and do not contribute to crosslinking activity. We denote the concentration of this PRC1 population by $c_f$ and assign no spatial dependence to it, since intracellular diffusion occurs at much faster time scales than metaphase and would therefore manage to equilibrate the free PRC1 population in the cell.

The singly bound population includes PRC1 molecules that are bound to a single microtubule only and, similar to the free population, do not contribute to crosslinking activity. We denote this population by $c_1(\vec{r})$ and relate it to the local tubulin concentration $\rho_{\mathrm{MT}}(\vec{r})$ via

$$c_1(\vec{r}) = \frac{\rho_{\mathrm{MT}}(\vec{r})}{K_d} c_f, \tag{26}$$

where $K_d$ is the dissociation constant of PRC1–single microtubule binding. To write the above relation between the free and singly bound populations, we again considered an equilibrated scenario, which we assume holds true given the fast dynamics of molecular turnover (*Pamula et al., 2019*) and diffusion compared to the duration of metaphase.

If $c_{\mathrm{tot}}(\vec{r})$ is the local concentration of all PRC1 populations together, then the doubly bound PRC1 population ($c_2(\vec{r})$) can be isolated by subtracting the free and singly bound populations from the total one, namely

$$\begin{aligned} c_2(\vec{r}) &= c_{\mathrm{tot}}(\vec{r}) - c_1(\vec{r}) - c_f \\ &= c_{\mathrm{tot}}(\vec{r}) - \frac{\rho_{\mathrm{MT}}(\vec{r})}{K_d} c_f - c_f. \end{aligned} \tag{27}$$

We are interested in estimating $c_2(\vec{r})$ along the pole-pole axis of the spindle in order to infer the length scale of the active crosslinking region.

To that end, for each spindle, we first estimate $c_f$ by averaging over the measured immunofluorescence in several different regions of interest (ROIs) where there is little to no detectable presence of microtubules. Examples of such ROIs are shown in *Appendix 5—figure 1a*. Next we need to estimate the dissociation constant $K_d$. Based on the in vitro measured $\approx$ 30-fold higher binding affinity of PRC1 to antiparallel microtubules compared to parallel ones (*Bieling et al., 2010*), and the result of an electron microscopy study suggesting that microtubules near the spindle poles are predominantly parallel (*Euteneuer and McIntosh, 1981*), we assume that the PRC1 population in the immediate vicinity of spindles poles is made out of free and singly bound states only. Denoting the pole-proximal positions by $\vec{r}_p$, we set $c_2(\vec{r}_p) \approx 0$ and use *Equation 27* to estimate $K_d$ as

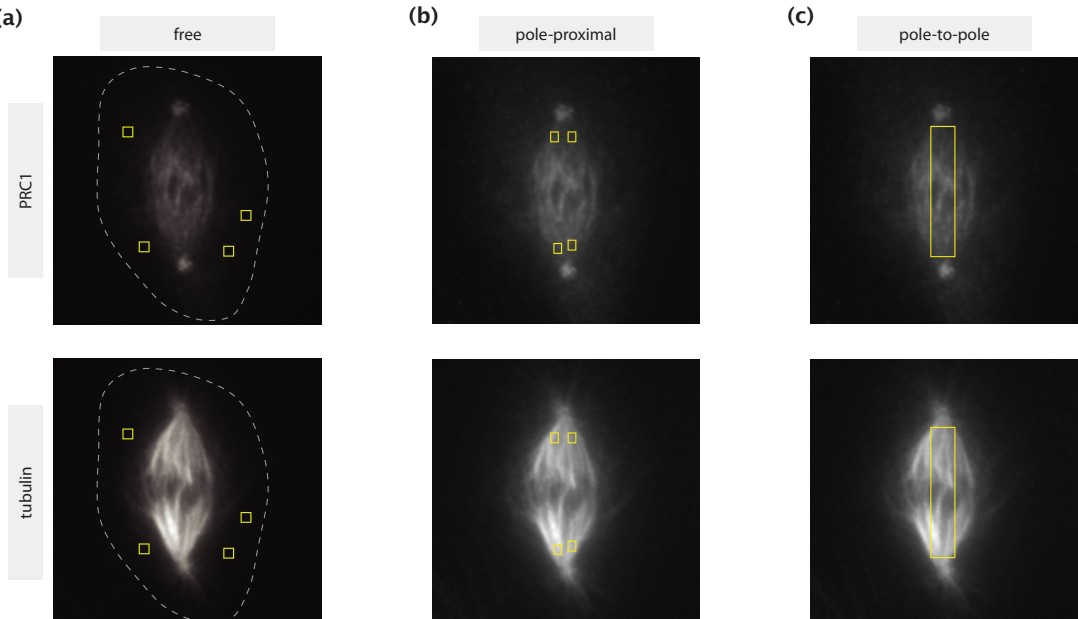

**Appendix 5—figure 1.** ROIs selected for different calculations shown on immunofluorescence images of PRC1 (top row) and tubulin (bottom row). (**a**) Regions with little to no tubulin presence where PRC1 can be considered unbound. The dashed lines represent the cell boundaries estimated by manual tracing based on high intensity contrast. (**b**) Pole-proximal regions where microtubules are present primarily in a parallel configuration. (**c**) Rectangular pole-to-pole region where the estimation of the actively engaged PRC1 population is made.

$$K_d \approx \left\langle \frac{\rho_{\mathrm{MT}}(\vec{r}_p)\, c_f}{c_{\mathrm{tot}}(\vec{r}_p) - c_f} \right\rangle_{\vec{r}_p},$$

(28)

where $\langle \cdot \rangle$ represents averaging over pole-proximal positions $\vec{r}_p$. Manually selecting several ROIs near the poles (*Appendix 5—figure 1b*) and using the immunofluorescence measurements for $c_{\mathrm{tot}}(\vec{r}_p)$ and $\rho_{\mathrm{MT}}(\vec{r}_p)$ in these regions, we perform the averaging and obtain the estimate for $K_d$.

With $c_f$ and $K_d$ calculated, we obtain the spatial profiles of actively engaged PRC1 molecules along the pole-pole axis of the spindle by selecting a rectangular region spanning the area between the poles (*Appendix 5—figure 1c*) and using the measured PRC1 ($c_{\mathrm{tot}}(\vec{r})$) and tubulin ($\rho_{\mathrm{MT}}(\vec{r})$) profiles to calculate $c_2(\vec{r})$ via *Equation 27*. Lastly, approximating k-fibers as homogeneous bundles of microtubules, we divide the calculated concentration of actively engaged PRC1 molecules by the local tubulin concentration, and report that ratio (engaged PRC1 per tubulin – a proxy for the strength of local crosslinking) as a function of position in the main text (*Figure 4i*).

