## [Editor Report]

In this elegant and technically sophisticated study, the authors study the mechanical properties of the mitotic spindle by combining various experimental biophysical approaches, including microneedle manipulation and quantitative imaging, with theoretical modeling. By systematically exploring shapes of unmanipulated and manipulated kinetochore fibers, they provide compelling evidence for a lateral anchor near the chromosomes. These important findings further our understanding of the balance of forces in the entire mitotic spindle. The work should appeal broadly to cell biologists and biophysicists who are interested in the cytoskeleton and cell division.

---

## [Decision Letter]

**Decision letter after peer review:**

Thank you for submitting your article "Modeling and mechanical perturbations reveal how spatially regulated anchorage gives rise to spatially distinct mechanics across the mammalian spindle" for consideration by *eLife*. Your article has been reviewed by 2 peer reviewers, and the evaluation has been overseen by a Reviewing Editor and Anna Akhmanova as the Senior Editor. The following individual involved in review of your submission has agreed to reveal their identity: Nenad Pavin (Reviewer #1).

Essential revisions:

1) Both reviewers noted that the model developed here seems to imply that kinetochores are compressed instead of being under tension as is well-established. This criticism either requires clarification or a modification of the model.

2) Moreover, the model assumes a uniform flexural rigidity of kinetochore fibers from pole to kinetochore. Could a higher flexibility close to the poles explain the observed shapes without invoking lateral enforcements near the kinetochores? This question should at least be discussed.

Other constructive comments that can be found below may help to improve the presentation of the manuscript.

*Reviewer #1 (Recommendations for the authors):*

(1) The authors analyze the shape of kinetochore fibers from two-dimensional images. However, the mitotic spindle is a three-dimensional object and the results could change if full three-dimensional shapes are taken into account. Please justify your approach, e.g. by showing a three-dimensional image of the spindle.

(2) The anchorage region is described by a series of elastic springs. These springs exert forces in y-direction, which is a valid approach in the small-angle approximation only. Please warn the reader about this approximation.

*Reviewer #2 (Recommendations for the authors):*

The authors have put much effort into trying to communicate sophisticated mechanical concepts to a broad audience, which is commendable, and they have created many diagrams to help readers understand. However, it remains difficult to follow certain aspects of the analyses. In particular, what are the directions and magnitudes of the end-applied forces predicted by the different models? Usually k-fibers in metaphase cells are considered to apply pulling forces, not pushing forces, to the kinetochores. The diagrams in Figure 1 (bottom), 2 a-b, and 3 b and d seem to suggest that the k-fibers support compressive/pushing forces, rather than pulling forces. If the models indeed predict compressive forces, this prediction should be explicitly stated along with some ideas about how to reconcile such forces with the prior evidence that kinetochores are usually under tension.

The magnitudes of the predicted forces would also be of interest. The very last data element of the main figures (Figure 5i) suggests that the lateral anchorage forces per k-fiber range from about 100 to about 600 pN. What are the predicted magnitudes for the forces at the bundle ends? How do these predicted forces compare with estimates of kinetochore force from other studies?

Line 206: "…although a moment at the kinetochore restricts free pivoting, it does so too locally, and thus fails to preserve k-fiber orientation over a few micrometers across the spindle center." Here is where I began wondering whether the assumption that flexural rigidity is uniform might break down. Could the k-fiber be more difficult to bend near the kinetochore than near the pole?

Line 228: "…we systematically tuned the length scale of lateral anchorage." Isn't it also necessary to tune the stiffness of these springs?

I don't fully understand why a model with distributed anchorage was not feasible, as mentioned on line 295: "Because of the challenges in extracting an accurate deformation map of the anchoring network under manipulation, using a model with distributed springs…was not feasible." It seems like one could in principle take the original shape of the k-fiber, prior to manipulation, and then arrange a set of springs between this original shape and the final deformed shape.

This phrase on line 334 seems vague, cumbersome: "…we sought to further dissect the results of model inference to learn about the emergent mechanics of the anchoring network dictating this response."

I do not understand this phrase starting on line 393: "This offers a potential explanation for why anchorage of this precise length scale can provide a robust connection to the dynamic spindle…" Perhaps it would help to explain what the authors think might occur as a chromosome oscillates through the region of lateral anchorage. Will lateral anchorage impede chromosome movements in this region? Will a chromosome "erase" lateral anchorage as it moves through the region? Will lateral anchorage allow motors (such as Kinesin-5s) acting on midzone microtubules to exert force indirectly onto the k-fibers and thus to help move the chromosomes poleward?

---

## [Author Response]

Essential revisions:1) Both reviewers noted that the model developed here seems to imply that kinetochores are compressed instead of being under tension as is well-established. This criticism either requires clarification or a modification of the model.

We thank the reviewers for their comment on this important point requiring

further clarification. Although the deformed native k-fiber is generally under compression for some of its length (Rubinstein et al. 2009, Kajtez et al. 2016), it is well established that the kinetochore is under tension (McNeill and Berns 1981, Waters et al. 1996). In our work, we sought to build a minimal model with the smallest set of effective parameters that would be sufficient to recapitulate native k-fiber shapes, and importantly, would allow us to build up model complexity to capture the anchorage revealed during manipulation. To this end, we coarse-grained the collective influence of compressive and tensile forces near the kinetochore to a single effective point force at the kinetochore-end. To address the reviewers’ comment, we performed a new study in which we justify our coarse-grained treatment and demonstrate that a fine-grained model in which the kinetochore-end force is tensile produces results that are consistent with our coarse-grained model. We describe this study below.

The bridging-fiber, a bundle of antiparallel microtubules that connects to the k-fiber, is thought to play a crucial role in balancing the tension and compression near the kinetochore (Kajtez et al. 2016). With this in mind, we built a fine-grained native k-fiber model in which we imposed the force at the kinetochore to be tensile and the junction force from the bridging fiber to be compressive (Figure 2 —figure supplement 1a). We then performed model fits to the subset of tracked k-fiber profiles where the junction positions were identifiable (Figure 2 —figure supplement 1b), and compared them to those of the original coarse-grained model. We observed a comparable quality of fits between the two models (example in Figure 2 —figure supplement 1c), and an overall good correspondence between the inferred force parameters at the pole (Figure 2 —figure supplement 1d). This suggests that using our coarse-grained model is justified for the purposes of native k-fiber shape studies (Figure 2a) conducted in our work. Notably, under external microneedle force, our manipulated coarse-grained k-fiber model (Figure 5a) infers the effective kinetochore force to be tensile, without explicitly including other force parameters.

In the revised manuscript, we present this study in Figure 2—figure supplement 1 and provide a detailed description in the methods section (lines 584-606) titled “Coarse-graining the kinetochore-proximal forces in native k-fibers”. We also clarify the use of our coarse-graining approach in the main text (lines 130-135): “In our minimal description, we coarse-grained the kinetochore-proximal forces a tensile force at the kinetochore (McNeill and Berns 1981, Waters et al. 1996) and a compressive force near the kinetochore (Rubinstein et al. 2009, Kajtez et al. 2016) to an effective point force (see Methods). Using a fine-grained junction model with an explicit tensile and compressive force did not significantly change the model outcomes (Figure 2 —figure supplement 1), thereby justifying our coarse-grained approach.”

2) Moreover, the model assumes a uniform flexural rigidity of kinetochore fibers from pole to kinetochore. Could a higher flexibility close to the poles explain the observed shapes without invoking lateral enforcements near the kinetochores? This question should at least be discussed.

We thank the reviewers for mentioning this interesting alternative mechanism for explaining the observed k-fiber response. To test this possibility, we performed extensive simulation studies for k-fibers with nonuniform flexural rigidity. Based on electron microscopy data (McDonald et al. 1992, O'Toole et al. 2020) we assumed in these new simulations that k-fibers have more microtubules and are more rigid in the kinetochore-proximal region. We found that while this mechanism could potentially shift the negative curvature minimum away from the kinetochore, for this shift to reach 2-3 µm away (as observed in the data, Figure 3h) the k-fiber would need to have more than twice as many microtubules near the kinetochore, which is far more than what electron microscopy studies indicate (20-30% more microtubules near the kinetochore) (McDonald et al. 1992, O'Toole et al. 2020). Thus, while a nonuniform flexural rigidity can shift the negative curvature minimum position, our model suggests that it cannot be the sole contributor to the observed manipulated k-fiber shapes.

We refer the reviewers to the newly added Figure 3 —figure supplement 2 and the associated Methods section (Lines 653-681) titled “Assumption of a uniform flexural rigidity” for details on this study. We also comment on this in the revised text (lines 212-219): “Motivated by electron microscopy studies demonstrating that k-fibers have 20-30% more microtubules near kinetochores compared to poles (McDonald et al. 1992, O'Toole et al. 2020), we also tested the impact of having a nonuniform k-fiber flexural rigidity on the position of the negative curvature minimum. We found that a local increase in flexural rigidity can shift the position of negative curvature away from the kinetochore (Figure 3 —figure supplement 2a-b). However, for the curvature minimum be up to 3 μm away from the kinetochore, the kinetochore-proximal region would need to have twice as many microtubules than the rest of the k-fiber (Figure 3 —figure supplement 2c-d), which is inconsistent with structural studies (McDonald et al. 1992, O'Toole et al. 2020)

Reviewer #1 (Recommendations for the authors):(1) The authors analyze the shape of kinetochore fibers from two-dimensional images. However, the mitotic spindle is a three-dimensional object and the results could change if full three-dimensional shapes are taken into account. Please justify your approach, e.g. by showing a three-dimensional image of the spindle.

We thank the reviewer for this point, which we clarify in our revised submission. For both our native and manipulated k-fiber datasets, we only included those k-fibers that were entirely visible within the same focal plane. This criterion allowed us to be able to track their shapes accurately and consistently. Since the shape information encoded in our dataset was effectively 2-dimensional, we employed a simple 2D model for k-fibers in our work. For future studies, expanding our model to three dimensions and considering the potential effects of torsional forces (Novak et al. 2018) on the k-fiber’s response to force would be an interesting future direction.

Currently, the methods section of our manuscript states “K-fibers were included in the data set only if their entire length stayed within the same z-plane over time, to enable accurate profile extraction” (lines 528-529). In the revised manuscript, we also added a discussion point stating that “… we only consider forces and moments that influence k-fiber shape in two dimensions. Looking forward, it will be useful to expand our model to include the potential effects of torsional forces on k-fiber shape generation (Novak et al. 2018)” (line 457-459).

(2) The anchorage region is described by a series of elastic springs. These springs exert forces in y-direction, which is a valid approach in the small-angle approximation only. Please warn the reader about this approximation.

As suggested by the reviewer, we have added a clarification in the Methods section describing our treatment of distributed anchorage: “We model this anchorage as a series of elastic springs exerting vertical pulling forces on the k-fiber. We ignore potential contributions in the horizontal direction which could have a non-negligible effect in the case of large network deformations” (lines 628-631).

Reviewer #2 (Recommendations for the authors):The authors have put much effort into trying to communicate sophisticated mechanical concepts to a broad audience, which is commendable, and they have created many diagrams to help readers understand. However, it remains difficult to follow certain aspects of the analyses. In particular, what are the directions and magnitudes of the end-applied forces predicted by the different models? Usually k-fibers in metaphase cells are considered to apply pulling forces, not pushing forces, to the kinetochores. The diagrams in Figure 1 (bottom), 2 a-b, and 3 b and d seem to suggest that the k-fibers support compressive/pushing forces, rather than pulling forces. If the models indeed predict compressive forces, this prediction should be explicitly stated along with some ideas about how to reconcile such forces with the prior evidence that kinetochores are usually under tension.

We thank the reviewer for their comment on this important point and refer them to our response on page 1 of this document (Essential Revisions).

The magnitudes of the predicted forces would also be of interest. The very last data element of the main figures (Figure 5i) suggests that the lateral anchorage forces per k-fiber range from about 100 to about 600 pN. What are the predicted magnitudes for the forces at the bundle ends? How do these predicted forces compare with estimates of kinetochore force from other studies?

We thank the reviewer for their comment on this important point. First, we refer the reviewer to our response to the editor’s first comment that addresses the concerns on the directionality of inferred kinetochore forces (on page 1 on this document). Second, to present our model inference results more clearly, we now include the inferred force arrows overlaid on example profiles for the native (Figure 2g and Figure 2 —figure supplement 2) and manipulated (Figure 5b) models.

The native model (Figure 2) infers end-forces (in the absence of an external microneedle force) to be on average 30 pN in magnitude, similar to what’s been inferred at the pole by other models in the literature (Rubinstein et al. 2009, Kajtez et al. 2016). We attempted to dissect the kinetochore forces for native k-fibers inferred from our junction model (in which the kinetochore-end is explicitly tensile) (Figure 2—figure supplement 2). However, because of the increased model complexity, the inferred parameters were degenerate (i.e. very different parameter combinations yielded similarly good fits to the data, as represented by the large error bars in Figure 2 —figure supplement 1d), making kinetochore-forces not identifiable. Lastly, we note that in response to external microneedle forces, the manipulated k-fiber model (Figure 5) infers the end-forces to be tensile and approximately an order of magnitude larger compared to the forces in the native state, at an average of 230 pN (Figure 5b).

Line 206: "…although a moment at the kinetochore restricts free pivoting, it does so too locally, and thus fails to preserve k-fiber orientation over a few micrometers across the spindle center." Here is where I began wondering whether the assumption that flexural rigidity is uniform might break down. Could the k-fiber be more difficult to bend near the kinetochore than near the pole?

We refer the review to our response on page 2 of this document.

Line 228: "…we systematically tuned the length scale of lateral anchorage." Isn't it also necessary to tune the stiffness of these springs?

We thank the reviewer for their comment. To define what spring stiffness could and could not contribute, we conducted an additional study where we varied the stiffness of the springs representing the anchorage network. We tested a range of values for anchorage strength spanning one order of magnitude from 0.2 and 2 nN/μm^2^, and tested its influence on the k-fiber’s negative curvature response near the kinetochore. For various choices of the anchorage length scale (σ), the tuning of anchorage strength did not significantly impact the positions of curvature minima. The largest stiffness-dependent variations in curvature minima positions were at most 1 μm when comparing the extreme cases (0.2 vs. 2 nN/μm^2^). While the precise choice of the anchorage strength may introduce small sub-micron variations in the negative curvature response, the length scale of anchorage (σ) remains the key determinant for where the minimum of the negative curvature is observed. In our revised manuscript, we have added this study in a new supplementary figure (Figure 4 —figure supplement 1) and a description in the main text:

“This conclusion holds true for a wide range of chosen anchorage strengths (Figure 4 —figure supplement 1).” (lines 259-260)

I don't fully understand why a model with distributed anchorage was not feasible, as mentioned on line 295: "Because of the challenges in extracting an accurate deformation map of the anchoring network under manipulation, using a model with distributed springs…was not feasible." It seems like one could in principle take the original shape of the k-fiber, prior to manipulation, and then arrange a set of springs between this original shape and the final deformed shape.

To infer stress (exerted forces) of the anchoring network, a detailed strain map of the anchoring network itself is needed (Sabass et al. 2008). However, our experimental and imaging system lacks the resolution in the anchoring network to do this accurately. Specifically, we lack information about the baseline state of the network before and after deformation, required for obtaining the strain map. One way to get around this could be to use the immediate neighboring k-fiber as a baseline, however, we cannot always see the neighboring k-fiber in the same focal plane during manipulation. A better way would be to introduce speckles in the anchoring network using fluorescent speckle microscopy, in order to directly visualize different regions of the network being deformed.

We worry that by using the initial and final shapes of the k-fiber (without information about the anchoring network’s baseline state), we would be making assumptions about how the network deforms, and this can have strong implications for the interpretation of the inferred forces. Additionally, a one-to-one mapping between the undeformed and deformed k-fiber is only possible if the k-fiber length remains constant throughout the manipulation. However, since our manipulations are performed over 1 min and k-fiber lengths change over this timescale, a simple one-to-one mapping is not possible. Therefore, our solution was to instead infer a single effective force near the kinetochore end, which captures the collective effects of the anchoring network stretching under manipulation force.

To better explain our rationale, we modify the text as follows:

"Because of the challenges in extracting an accurate strain map of the anchoring network and knowledge of the baseline state before and after manipulation, using a model with distributed springs (Figure 4a) that would require these as input information, was not feasible. Further, a simple one-to-one mapping between the undeformed and deformed k-fibers could not be done due to the k-fiber length changing over the 60-second manipulation.” (lines 311-315)

This phrase on line 334 seems vague, cumbersome: "…we sought to further dissect the results of model inference to learn about the emergent mechanics of the anchoring network dictating this response."

In the revised manuscript, we have simplified the phrasing of this sentence, which we hope brings more clarity to the ideas. Lines 353-354: “…we dissected the results of model inference – to investigate how the anchoring network responds to microneedle forces”

I do not understand this phrase starting on line 393: "This offers a potential explanation for why anchorage of this precise length scale can provide a robust connection to the dynamic spindle…" Perhaps it would help to explain what the authors think might occur as a chromosome oscillates through the region of lateral anchorage. Will lateral anchorage impede chromosome movements in this region? Will a chromosome "erase" lateral anchorage as it moves through the region? Will lateral anchorage allow motors (such as Kinesin-5s) acting on midzone microtubules to exert force indirectly onto the k-fibers and thus to help move the chromosomes poleward?

We have modified the sentences to improve clarity. The sentences now read as follows:

“Further, the presence of lateral anchorage across the center of the spindle can constrain chromosome oscillations, while still allowing movement on a longer timescale and ensuring sister k-fiber alignment. This together offers a potential explanation for why anchorage of this precise length scale can provide a robust connection to the dynamic spindle, and raises the question of how this length scale varies across spindles with different metaphase chromosome movement amplitudes” (lines 410-415).

References:

Kajtez, J, Solomatina, A, Novak, M, Polak, B, Vukusic, K, Rudiger, J, Cojoc, G, Milas, A, Sumanovac Sestak, I, Risteski, P, Tavano, F, Klemm, AH, Roscioli, E, Welburn, J, Cimini, D, Gluncic, M, Pavin, N, Tolic, IM. 2016. Overlap microtubules link sister k-fibres and balance the forces on bi-oriented kinetochores. Nat Commun 7: 10298.

McDonald, KL, O'Toole, ET, Mastronarde, DN, McIntosh, JR. 1992. Kinetochore microtubules in ptk cells. J Cell Biol 118(2): 369-383.

McNeill, PA, Berns, MW. 1981. Chromosome behavior after laser microirradiation of a single kinetochore in mitotic ptk2 cells. J Cell Biol 88(3): 543-553.

Novak, M, Polak, B, Simunic, J, Boban, Z, Kuzmic, B, Thomae, AW, Tolic, IM, Pavin, N. 2018. The mitotic spindle is chiral due to torques within microtubule bundles. Nat Commun 9(1): 3571.

O'Toole, E, Morphew, M, McIntosh, JR. 2020. Electron tomography reveals aspects of spindle structure important for mechanical stability at metaphase. Mol Biol Cell 31(3): 184-195.

Rubinstein, B, Larripa, K, Sommi, P, Mogilner, A. 2009. The elasticity of motor-microtubule bundles and shape of the mitotic spindle. Phys Biol 6(1): 016005.

Sabass, B, Gardel, ML, Waterman, CM, Schwarz, US. 2008. High resolution traction force microscopy based on experimental and computational advances. Biophys J 94(1): 207-220.

Waters, JC, Skibbens, RV, Salmon, ED. 1996. Oscillating mitotic newt lung cell kinetochores are, on average, under tension and rarely push. J Cell Sci 109 ( Pt 12): 2823-2831.